# Quantifying the effects of the microphysical properties of black carbon on the determination of brown carbon using measurements at multiple wavelengths

Jie Luo[1], Dan Li[1], Yuanyuan Wang[1], Dandan Sun[1], Weizhen Hou[3], Jinghe Ren[1], Hailing Wu[3], Peng Zhou[4], and Jibing Qiu[1,2,*]

[1]Zhejiang Lab, Hangzhou, Zhejiang 311121, China.
[2]Institute of Computing Technology, Chinese Academy of Sciences, Beijing 100190, China
[3]State Environment Protection Key Laboratory of Satellite Remote Sensing, Aerospace Information Research Institute, Chinese Academy of Sciences, Beijing 100101, China
[4]School of Surveying and Land Information Engineering, Henan Polytechnic University

**Correspondence:** Jibing Qiu (qiujibing@ict.ac.cn)

**Abstract.** The absorption Ångström exponent (AAE)-based methods are widely used to estimate brown carbon (BrC) absorption, and the estimated BrC absorption can be significantly different from 0 even for pure black carbon (BC). However, few studies have systematically quantified the effects of BC microphysical properties. Moreover, it is still unclear under which conditions the AAE-based method is applicable. In this work, we used BC models partially coated with non-absorbing materials to calculate the total absorption. Since the total absorption is entirely from BC, the estimated BrC absorption should be 0 if the retrieval methods are accurate. Thus, the ratio of the estimated BrC absorption to BC absorption ($ABS_{BrC}$) should be the proportions of BC absorption that is incorrectly attributed to BrC. The results show that a BC AAE of 1 can generally provide reasonable estimates for freshly emitted BC, since at this time $ABS_{BrC}$ is generally in the range of -4.8% to 2.7%. However, when BC aerosols are aged, $ABS_{BrC}$ can sometimes reach about 38.7%. The wavelength dependence of AAE (WDA) method does not necessarily improve the estimates, sometimes a negative $ABS_{BrC}$ of about -40.8% can be found for partially coated BC. By combining simulations of a global chemical transport model, this work also quantified the effects of BC microphysical properties on BrC global aerosol absorption optical depth (AAOD) estimates. The AAE = 1 method could sometimes lead to a mis-assigned global mean AAOD of about $-0.43 – 0.46\times 10^{-3}$ if BC aerosols have a complex morphology. The WDA method does not necessarily improve the estimates. In our cases, the WDA methods based on the spherical models can lead to a global-mean mis-assigned AAOD range of about $-0.87 – 0.04 \times 10^{-3}$. At the regional scale, the AAE = 1 method in some specific regions sometimes leads to a distributed AAOD of about $-7.3$ to $5.7 \times 10^{-3}$. Mie theory-based WDA methods would lead to an estimated AAOD error of absout $-22 \times 10^{-3}$ in some regions (e.g., East Asia). This work also show that the mis-attributed BrC absorption would lead to substantial uncertainties in the estimation of global direct radiative forcing (DRF) of absorbing aerosols from different sources.

# 1 Introduction

Carbonaceous aerosols are a major contributor to climate change (Bond et al., 2013; Myhre et al., 2013). Black carbon (BC) and organic carbon (OC) are the most important carbonaceous aerosols in the atmosphere. As the most important absorbing aerosol in the atmosphere, BC significantly absorbs solar radiation from ultraviolet (UV) to near-infrared (NIR) wavelengths (Cross et al., 2010; Petzold and Schönlinner, 2004; Bond and Bergstrom, 2006). Even at small mass fractions of total atmospheric aerosols, the radiative effects of BC are quite significant due to high absorption, which greatly enhances global warming (Zhang and Wang, 2011; Bond et al., 2013; Matsui et al., 2018). In contrast, OC was initially considered as a mere scattering agent that has a cooling effect on the Earth-atmosphere system (Wang et al., 2016). However, recent studies have shown that some OC aerosols also absorb light from UV to visible wavelengths (Kirchstetter et al., 2004; Chakrabarty et al., 2010; Chen and Bond, 2010). These OC aerosols are known as brown carbon (BrC) (Andreae and Gelencsér, 2006; Laskin et al., 2015). Recent studies have also shown that BrC also exerts considerable positive radiative forcing (Zeng et al., 2020b; Feng et al., 2013). In some specific regions, the direct radiative effect of BrC is even comparable to that of BC (Zhang et al., 2020a). Therefore, understanding BrC absorption is important for studying global climate change.

However, the understanding of BrC absorption is still quite limited. BrC strongly absorbs light in the UV region, while its absorption strongly decreases with increasing wavelength from the UV region to the visible region (Hecobian et al., 2010; Kirchstetter et al., 2004; Chakrabarty et al., 2010; Chen and Bond, 2010; Bahadur et al., 2012). Filter samples in laboratory measurements are the main method for measuring the absorption characteristics of BrC (Chen and Bond, 2010; Xie et al., 2019). Based on laboratory studies, previous studies have shown that the mass cross section (MAC) of BrC varies over a wide range and that the reported values in different studies are different (Dasari et al., 2019; Kirillova et al., 2016). The reasons for the large uncertainties are mainly due to the samples from different regions and different measurement conditions. Compared with BC, the absorption of BrC is subject to larger uncertainties due to the type of fuel, combustion condition, aging condition, etc. Therefore, it is difficult to make accurate estimates of BrC absorption based on laboratory measurements in a global range because it is unrealistic to perform filter sampling under all conditions. Numerical modeling is another effective tool for estimating BrC absorption. However, modeling is usually based on prior knowledge from laboratory measurements, which varies by region, weather conditions, aging condition, etc. Therefore, even with modeling, it is difficult to always provide accurate estimates.

Remote sensing can provide regional/global measurements and is an effective complementary method to address the above issues. Recently, researchers have attempted to derive the absorption of BrC based on the absorption of multiple wavelengths from remote sensing (Arola et al., 2011; Tesche et al., 2011; Wang et al., 2016). Ground-based remote sensing, such as the Aerosol Robotic Network (AERONET) (Holben et al., 1998; Giles et al., 2019), could provide measurements of the temporal aerosol absorption optical depth (AAOD) on a global scale (Shaw, 1983; Shin et al., 2019). In addition, polarimetric satellite measurements have also been used to determine AAOD (Dubovik et al., 2011, 2014). With the increasing number of satellites, it is expected that the absorbing aerosols on a global scale can be detected in real time in the future through the cooperation of satellite constellations. However, the absorption derived from remote sensing is the mixture of different absorbing aerosols.

Dust, BrC, and BC are the main absorbing aerosols in the atmosphere, and we usually need to separate the contributions of dust and BC to study the absorption of BrC. Since dust particles are usually aerosols of large size, we can break down the contribution of dust based on the size information. Specifically, scattering/Extinction Ångström exponent could be used to infer the size information, and they were used in previous studies to seperate dust from BC and BrC (Cazorla et al., 2013; Russell et al., 2010; Cappa et al., 2016). However, both BC and BrC are fine aerosols, and it is difficult to separate them based on size information alone.

An effective technique for separating the contributions of BC is based on the different spectral absorption dependence of BC and BrC (Wang et al., 2016, 2018; Russell et al., 2010; Chung et al., 2012). The absorption of BrC is generally very weak at long visible wavelengths and in the NIR, and it has been generally assumed that the total absorption is entirely from BC (Wang et al., 2016; Luo et al., 2021c). Since the spectral dependence of the absorption of BC is subject to relatively small uncertainties, the absorption of BC in the UV region can be derived from the absorption in the long visible and NIR regions. Then, the absorption of BrC in the UV region could be estimated from the difference between the total absorption and the BC absorption. At two reference wavelengths ($\lambda_1$, $\lambda_2$), the spectral dependence of absorption is generally represented by a parameter, the absorption Ångström exponent (AAE):

$$AAE = -\frac{\ln(b_{abs(\lambda_1)}) - \ln(b_{abs(\lambda_2)})}{\ln(\lambda_1) - \ln(\lambda_2)} \tag{1}$$

where $b_{abs(\lambda_1)}$ and $b_{abs(\lambda_2)}$ are the absorption coefficients at $\lambda_1$ and $\lambda_2$, respectively.

Using the BC AAE value, the absorption of BC in the UV region can be estimated based on the absorption at NIR wavelengths. When estimating BrC absorption, a BC AAE of 1 has often been used, while more recent studies based on measurements and simulations have shown a wide range of AAE values (Schnaiter et al., 2005; Lack and Cappa, 2010b; Li et al., 2016; Liu et al., 2018). Therefore, the use of BC AAE of 1 may not always provide accurate estimates. To correct the AAE = 1 method, Wang et al. (2016) proposed to use the wavelength dependence of AAE (WDA) to improve the estimates. However, in their studies, the WDA was calculated using the Mie theory, which assumes that the morphology of BC is spherical. However, in the atmosphere, BC exhibits a rather complex morphology. Luo et al. (2021c) have shown that the WDA method does not necessarily lead to better estimates when BC has a complex morphology. Thus, the estimation of BrC based on absorption measurements at multiple wavelengths suffers from uncertainties in BC properties.

It is well known that the AAE method does not provide accurate results. However, there is a lack of understanding of the uncertainties caused by the microphysical properties of BC. Using morphologically realistic models, Luo et al. (2021c) showed that the estimates of BrC-based AAE methods are significantly affected by BC morphology, while how the uncertainties are affected by BC microphysical properties was not investigated. Moreover, in their work only single particles are considered, while bulk particles exist in the atmosphere. In this work, we attempt to systematically quantify the impact of BC microphysical properties on the estimates of BrC-based AAE methods. We focus on answering the following questions:

– How large are the uncertainties caused by the microphysical properties of BC in the estimation of BrC?

– How do the microphysical properties of BC affect estimates of BrC?

– What implications can we obtain for estimates of BrC on a global scale?

To answer the above questions, we generated some "realistic" BC aerosols based on the partially coated models and then estimated the absorption fraction of BC incorrectly attributed to BrC ($ABS_{BrC}$) using different AAE methods and investigated the effects of BC microphysical properties on $ABS_{BrC}$. Finally, assuming a typical size distribution, we investigated the global distribution of BC AAOD the mis-attributed to BrC at different BC morphologies and mixing states based on a global chemical transport model, the Goddard Earth Observing System with chemistry (GEOS-Chem). Our study can improve the understanding of uncertainties in the estimation of BrC based on absorption measurements at multiple wavelengths.

## 2 Estimating the BrC absorption

### 2.1 BC morphological model

In calculating the optical properties, the shape of BC was largely assumed to be spherical, so we can calculate the optical properties using Mie theory (Mie, 1908; Lack and Cappa, 2010a; Bao et al., 2019). However, BC in the atmosphere usually has a rather complex morphology (China et al., 2015; Adachi et al., 2010; Yuan et al., 2019; Wang et al., 2017; Luo et al., 2021a). When BC is freshly emitted, the morphology is usually chain-like and consists of numerous spherical particles (Sorensen, 2001, 2011). Researchers have often used the fractal law to describe the morphology (Sorensen, 2001; Heinson et al., 2017; Luo et al., 2021a, b):

$$N_s = k_0 \left(\frac{R_g}{R}\right)^{D_f} \tag{2}$$

where $N_s$ is the number of spherical monomers; $k_0$ and $D_f$ are two parameters representing the symmetry and compactness of the BC aggregates and are called fractal prefactor and fractal dimension, respectively; $R$ and $R_g$ represent the radius of the small particles and the gyration radius, respectively.

$D_f$ is often used to describe the compactness of BC (Liu and Mishchenko, 2005; Radney et al., 2014). BC generally exhibits a fluffy morphology when emitted into the atmosphere. Previous studies have shown that a small $D_f$ can strongly affect the morphology of BC. For BC aerosols from biomass combustion, a $D_f$ range of 1.67 – 1.83 was observed (Chakrabarty et al., 2006) ; BC aerosols from vehicle emissions showed a $D_f$ range of 1.52 – 1.94 (China et al., 2014); Wentzel et al. (2003) showed a $D_f$ of 1.6 – 1.9 for fresh BC from vehicle emissions. From a robotics perspective, previous studies have shown that simulations based on the diffusion-limited cluster-cluster aggregation (DLCA) algorithm can explain the measurements well, and a universal $D_f$ of about 1.8 was obtained by the simulations (Sorensen, 2001; Dhaubhadel et al., 2006). Therefore, we used a $D_f$ of 1.8 to represent the fluffy BC.

With atmospheric aging, BC can be reconstructed into a condensed structure (Lack et al., 2014; Zhang et al., 2008; Bhandari et al., 2019). In many studies, a larger $D_f$ was used to represent the compact BC (Liu et al., 2017; Luo et al., 2018, 2019). Previous studies have shown that the $D_f$ of aged BC can sometimes reach about 2.3– 2.6 (Adachi et al., 2010, 2007; Chen et al., 2016), and we used a $D_f$ of 2.6 to represent compact BC. At the same time, the surface of black carbon is covered with other

materials, which makes the morphology more complex (China et al., 2013; Pang et al., 2023, 2022; Wang et al., 2021). In the atmosphere, some BC cores are fully coated while others are partially coated. Both the fluffy BC and the compact BC may be partially coated, while the fully coated BC generally has a compact BC core due to the condensation of the coating materials. Therefore, we assumed the following cases for aged BC aerosols: (1) fluffy BC cores partially coated with other materials; (2) compact BC without coating materials; (3) compact BC partially coated with other materials; (4) compact BC fully coated with other materials. It should be noted that BC with thicker coating materials become more easily compact, so we assumed a $D_f$ of 2.6 for the fully coated BC.

The DLCA algorithm was developed to characterize the morphology of fresh BC, and the aggregates generated generally have a $D_f$ of about 1.8 and a $k_0$ of about 1.3 (Sorensen and Roberts, 1997; Heinson et al., 2010, 2017, 2018). However, the DLCA algorithm is not able to characterize the morpology of the aged BC with a compact structure. Since the fractal structure is well described by equation (2), tunable algorithms based on equation (2) were developed to replace DLCA. Compared to the DLCA algorithm, the tunable algorithms are parametrically adjustable ($k_0$ and $D_f$ are fully adjustable) (Filippov et al., 2000; Skorupski et al., 2014; Moran et al., 2019). Although the tunable algorithm does not provide physical explanations of how the morphologies are generated exactly like DLCA, it can represent the DLCA aggregate by setting $k_0 = 1.3$ and $D_f = 1.8$. Moreover, the tunable algorithm can represent more compact BC with larger $D_f$. Therefore, we used a tunable algorithm developed by Woźniak (2012) to generate the uncoated BC cores.

After the BC cores were generated, we added the coating materials on the surface of the BC cores. Similar to previous studies (Zhang et al., 2018; Luo et al., 2018; Liu et al., 2017), we adopted a spherical coating structure for coated BC. It is noted that coating materials much more complex than the spherical structure can be observed in the atmosphere. However, as previous studies have shown, the absorption of coated BC is significantly affected by the "lensing effect", which largely depends on the fraction of coated BC cores. Compared with the fraction of BC cores in the coating materials (F), the effects of the coating structures on the absorption of BC are relatively small. Therefore, we only consider the F with a partially coated model with a spherical coating structure, and the complex coating structures were not considered in this work. The effects of coating structures have been studied elsewhere (Luo et al., 2019).

The spherical coating materials were generated similarly to Zhang et al. (2018). F is calculated using the following equation:

$$F = \frac{V_{BC\ inside}}{V_{BC}} \tag{3}$$

where $V_{BC\ inside}$ represents the BC volume inside the coating shell, and $V_{BC}$ represent the volume of the total coated BC.

Once the BC volume fraction ($f_{BC}$) and F are given, we can determine the radius of the spherical coating. Then the BC cores are moved from side to side until the given F is reached, similar to Zhang et al. (2018). It should be noted that the motion would lead to an overlap of the BC core and the coating sphere, and their optical properties cannot be calculated by some efficient numerical methods, such as the multiple sphere T-matrix (MSTM) method (Mackowski and Mishchenko, 2011; Mackowski, 2022), which is only applicable for multiple spheres without overlap. For efficient calculations, we moved the overlapping BC sphere cores outside the coating sphere, similar to Zhang et al. (2018), and the movement would not affect

the optical properties of coated BC significantly (Liu et al., 2017). The typical BC morphologies are shown in Figure 1. Since

we consider only spherical coating structures in this work, a large F for completely BC may not be found for a BC volume

fraction. Therefore, we only consider an F range from 0 to 0.3 for fluffy BC. However, we assume that BC with large F would

not change the main results of this work.

## 2.2   The calculations of BC absorption

Similar to our previous study (Luo et al., 2018), the absorption cross sections ($C_{abs}$) of individual BC particles were calculated

by the MSTM method, which can be used to calculate the optical properties of spheres without overlap. It has been reported

that the refractive index of BC varies with wavelength (Chang and Charalampopoulos, 1990), while it is generally known that

it does not vary significantly from visible to near-infrared wavelength (Bond and Bergstrom, 2006). Therefore, a fixed BC

refractive index of $1.95 + 0.79i$ was assumed according to the suggestions of Bond and Bergstrom (2006). BrC aerosols are

organic carbon that absorbs light from UV to visible wavelengths. Therefore, the coating shell was assumed to be organic

carbon, and the refractive index was taken to be 1.55 (Bond and Bergstrom, 2006). It should be noted that the imaginary part of

BrC should not be 0 in principle. However, in this work we are mainly concerned with studying the absorption of nonabsorbing

aerosols mixed with BC, which are erroneously attributed to BrC, so we considered only one nonabsorbing shell. The details

are presented in the following sections.

Once the BC structures were generated and the refractive index was specified, the MSTM was used to calculate the absorption

of each BC particle. The MSTM directly outputs an effective absorption efficiency ($Q_{abs}$), which is defined in terms of the

volume mean radius. The absorption cross section ($C_{abs}$) can be calculated as:

$$C_{abs} = Q_{abs}\pi r_v^2 \tag{4}$$

where $r_v$ represents the mean volume radius.

In the atmosphere, numerous BC aerosols exist, and the optical properties of BC should average over all the particles. Thus,

we calculated the bulk optical properties by assuming different size distributions. Assuming a lognormal distribution for the

size distribution of BC cores:

$$n(r_v) = \frac{1}{\sqrt{2\pi}r_v\ln(\sigma_g)}\exp\left[-\left(\frac{\ln(r_v) - \ln(r_g)}{\sqrt{2}\ln(\sigma_g)}\right)^2\right] \tag{5}$$

where $r_g$ and $\sigma_g$ are the geometric mean radius and geometric standard deviation, respectively.

We first calculated the $C_{abs}$ of BC with different $N_s$ (i.e., different $r_v$), and then calculated the absorption coefficient ($b_{abs}$)

using:

$$b_{abs} = \int_{r_{min}}^{r_{max}} C_{cabs}(r_v)n(r_v)dr_v \tag{6}$$

where $r_{max}$ and $r_{min}$ are the maximum radius and minimum radius of BC cores, respectively.

The $r_g$ and $\sigma_g$ values reported in this work are for the BC cores and can be determined using some single-particle technologies, such as the soot particle photodiameter (SP$_2$) (Moteki et al., 2007; Baumgardner et al., 2004; Schwarz et al., 2006). Many measurements were based on single particle technologies, and different $r_g$ and $\sigma_g$ were observed in different regions. BC particles in Tokyo were observed with a geometric mean radius ($r_g$) of about $0.032 \pm 0.003$ $\mu$m and a geometric standard deviation ($\sigma_g$) of $1.66 \pm 0.12$ (Kondo et al., 2011). However, Shiraiwa et al. (2008) showed that $r_g$ and $\sigma_g$ of BC in Fukue, Japan, are $0.095 - 0.105$ $\mu$m and $1.45 - 1.55$, respectively. At BC in Shanghai, $r_g$ of about 0.1 $\mu$m (Gong et al., 2016) was observed. In general, small BC particles are easier to be fully coated, so partially coated BC usually has a large size. Therefore, we mainly calculated the optical properties of relatively large BC particles. $N_s$ of $5 - 1000$ were considered, and the corresponding $r_{min}$ and $r_{max}$ are $0.0342$ $\mu$m and $0.2$ $\mu$m. An $r_g$ of 0.05-0.1 $\mu$m was assumed, and the $\sigma_g$ was assumed to be in the range of 1.3 to 2.0.

## 2.3  Inferring the fractions of BrC absorption

We estimated the absorption of BC, which was incorrectly attributed to BrC, after calculating BC absorption. In AERONET, the 440 nm, 675 nm, and 870 nm wavelengths were most commonly used to estimate BrC absorption, and we mainly considered these three wavelengths. At 675 nm and 870 nm, all of the absorption was assumed to come entirely from BC. The $b_{abs}$ of BC at 440 nm can be determined as follows:

$$b_{abs\_BC\_440\_Estimated} = b_{abs\_BC\_\lambda}(\frac{440}{\lambda})^{-AAE_{\lambda\_440}} \tag{7}$$

where $AAE_{\lambda\_440}$ represents the AAE of BC for the $\lambda$ and 440 nm wavelength pair; $\lambda$ represents 870 nm or 675 nm wavelength.

The $b_{abs}$ of BrC at 440 nm can be estimated using:

$$b_{abs\_BrC\_440\_Estimated} = b_{abs\_440\_total} - b_{abs\_BC\_440\_Estimated} = b_{abs\_BC\_440} + b_{abs\_BrC\_440} - b_{abs\_BC\_440\_Estimated} \tag{8}$$

where $b_{abs\_BrC\_440\_Estimated}$ and $b_{abs\_440\_total}$ represent the estimated BrC absorption coefficient and total absorption coefficient at 440 nm, respectively. $b_{abs\_BC\_440}$ and $b_{abs\_BrC\_440}$ represent the "true" BC and BrC absorption coefficients at 440 nm, respectively. Thus, the absorption coefficient of BC, which is incorrectly attributed to BrC, can be calculated as follows:

$$\Delta_{BrC} = b_{abs\_BrC\_440} - b_{abs\_BrC\_440\_Estimated} = b_{abs\_BC\_440\_Estimated} - b_{abs\_BC\_440} \tag{9}$$

Then we calculated the proportions of the wrongly assigned absorption with:

$$ABS_{BrC} = \frac{\Delta_{BrC}}{b_{abs\_BC\_440}} = \frac{b_{abs\_BC\_440\_Estimated}}{b_{abs\_BC\_440}} - 1 \tag{10}$$

The WDA method is similar to the fixed AAE methods, but the AAE are inferred from the Mie thoery:

$$WDA = AAE_{440\_870\_Mie} - AAE_{675\_870\_Mie} \tag{11}$$

where the subscript "Mie" stands for the AAE calculated using the Mie theory. Thus, we can estimate the $AAE_{440\_870}$ based on the WDA:

$$AAE_{440\_870} = AAE_{675\_870} + WDA \tag{12}$$

The $ABS_{BrC}$ in this work is retrieved from the absorption of BC. Consequently, any deviations from $ABS_{BrC} = 0$ suggest the uncertainty in BrC estimation. This work is based solely on modeling, and no measurements are used.

## 2.4 Calculating the global BC absorption

We used a global atmospheric chemical transport model, GEOS-Chem (Bey et al., 2001; Eastham et al., 2018), to simulate the global distribution of BC. For this work, GEOS-Chem version 12.7 was used with a latitude/longitude grid resolution of $4°\times 5°$. MERRA-2 (second Modern-Era Retrospective analysis for Research and Applications) assimilated meteorology(Molod et al., 2015; Gelaro et al., 2017) was used for the GEOS-Chem model simulations. The model was built with 47 vertical layers. We ran a standard GEOS-Chem simulation with various aerosols such as dust, organic aerosols, BC, sulphate, sea salt, etc. The Community Emissions Data System (CEDS) (Hoesly et al., 2018) inventory provided the global anthropogenic emissions. The Global Fire Emissions Database (GFED4) inventory (Randerson et al., 2018) was used to provide emissions from biomass burning. Biogenic emissions were obtained from the Model of Emissions of Gases and Aerosols from Nature Version 2.1 (MEGAN 2.1)(Guenther et al., 2012). We used GEOS-Chem to simulate black carbon concentrations for all of 2016, and then took the time average.

The BC AAOD in each GEOS-Chem grid, was calculated using:

$$AAOD_{BC} = MAC_{BC} \times C_{BC\_column} \tag{13}$$

where $C_{BC\_column}$ is the column mass concentrations of BC; $MAC_{BC}$ represents the mass cross sections of coated BC that can be calculated with:

$$MAC_{BC} = \frac{b_{abs}}{m_{BC}} \tag{14}$$

where $m_{BC}$ represent the mass distributions of BC cores which can be calculated with:

$$m_{BC} = \int_{r_{min}}^{r_{max}} \frac{\rho_{BC} 4\pi r_v^3 n(r_v)}{3} dr_v \tag{15}$$

where $\rho_{BC}$ represents the mass density of BC. Bond and Bergstrom (2006) to use a $\rho_{BC}$ of $1.8 \, \text{g m}^{-3}$. However, most modeling studies underestimated the proposed MAC of $7.5 \pm 1.2 \, \text{m}^2 \, \text{g}^{-1}$ at 550 nm using the proposed $\rho_{BC}$. Similar to Luo et al. (2018), we used a $\rho_{BC}$ of 1.5 to fit the MAC measurements.

The global AAOD of BC, incorrectly attributed to BrC, can be determined using the following method:

$$AAOD_{BrC} = AAOD_{BC} \times ABS_{BrC} \tag{16}$$

In addition to AAOD, the Direct Radiative Forcing (DRF) is also commonly used to assess climate effects. In this work, DRF is also estimated using a simple method. Based on the values in Bond et al. (2013), Kelesidis et al. (2022) proposed to use an average absorption forcing efficiency of $170 \pm 43$ Wm$^{-2}$/AAOD. Similar to Kelesidis et al. (2022), the DRF is estimated by multiplying the estimated AAOD by $170 \pm 43$ Wm$^{-2}$/AAOD. Similarly, the mis-assigned DRF is also estimated by multiplying the mis-assigned AAOD by $170 \pm 43$ Wm$^{-2}$/AAOD.

## 3    Results

### 3.1    The effects of BC microphysical properties on the fixed AAE method

Figure 2 shows the effects of the shell diameter to core diameter ratio ($D_p/D_c$). The error bars in the figures represent the upper and lower limits when $r_g$ is varied in the range of $0.05 - 0.1$ $\mu$m and $\sigma_g$ is varied in the range of $1.5 - 1.8$. Since BC aerosols are freshly emitted, they are fluffy and not mixed with the coating materials. Our cases where $D_f = 1.8$ and $F = 0$ may reflect freshly emitted BC. Previous studies have shown that the AAE of freshly emitted BC is not significantly different from 1 (Liu et al., 2018; Luo et al., 2020). Thus, using AAE$_{440\_675}$ = 1 and AAE$_{440\_870}$ = 1 can provide reasonable estimates of BrC absorption for freshly emitted BC, and the estimated ABS$_{BrC}$ is not significantly different from 0 when $D_p/D_c$ < 2.15. As shown in Table Based on different AAE wavelength pairs, the estimated ABS$_{BrC}$ shows some differences. The AAE$_{440\_870}$ = 1 method generally shows a larger range of ABS$_{BrC}$ than the AAE$_{440\_675}$ = 1. This could be due to the larger wavelength gap between 440 and 870 nm. ABS$_{BrC}$ ranges from about -2.8% to 2.5% when $D_p/D_c$ < is 2.15 and the AAE$_{440\_675}$ = 1 method is used, while this range becomes about -4.8% – 2.7% when the AAE$_{440\_870}$ = 1 method is used. As the number of coating materials increases, BC AAE may gradually deviate from 1 due to the shielding effect of heavy coatings. Thus, a broader ABS$_{BrC}$ of about -16.3% – -5.2% is obtained when $D_p/D_c$ increases to 4.64 using the AAE$_{440\_675}$ = 1 method. A relatively wider range was also observed when AAE$_{440\_870}$ = 1. ABS$_{BrC}$ ranges from about -32% to -6.9% when $D_p/D_c$ = 4.64 using the AAE$_{440\_870}$ = 1 method.

The bare, fluffy BC aggregates are gradually coated by other materials with atmospheric aging, so BC with a larger F represents more aged particles. Since the BC aggregates are partially coated, ABS$_{BrC}$ also gradually deviates from 0 for the fluffy BC aggregates. As shown in Figure 2 and Table 1, the increase of F can lead to a wider range of ABS$_{BrC}$. With $D_f$ of 1.8, F of 0.1, and $D_p/D_c$ of less than 2.15, ABS$_{BrC}$ varies in the range of about -6.3% – 10.4% and -11.2% – 9% when AAE$_{440\_675}$ = 1 and AAE$_{440\_870}$ = 1. The ranges become -18% – -4.71% and -22.8% – -6.9% when $D_p/D_c$ is 4.64. The ABS$_{BrC}$ range becomes larger as F increases to 0.3. An ABS$_{BrC}$ range of -34.5% – 21.2% can be found when BC with a $D_f$ of 1.8 and an F of 0.3 when assuming an AAE of 1. When BC aggregates are partially coated, the ABS$_{BrC}$ are generally smaller than 0 when $D_p/D_c$ is larger (e.g., $D_p/D_c$ = 4.64) when BC has a fluffy structure, whereas ABS$_{BrC}$ can be larger than 0 when $D_p/D_c$ is small. This phenomenon may be due to the different effects of coating ratios on AAE. When $D_p/D_c$ is small, the AAE is generally small and can sometimes be less than 1 (Zhang et al., 2020b; Luo et al., 2023; Liu et al., 2018). Therefore, assuming an AAE of 1 may overestimate the real AAE, so an ABS$_{BrC}$ greater than 0 can be observed. On the other hand, the AAE increases with $D_p/D_c$ when BC has a fluffy structure. Thus, the AAE can be greater than 1 when the fluffy BC is

partially coated with a thick coating (Zhang et al., 2020b; Luo et al., 2023). This would result in the predicted BC absorption

coefficient being greater than the actual BC absorption coefficient, resulting in an $ABS_{BrC}$ of less than 0. For a $D_f$ of 1.8, Luo

et al. (2023) have shown that the $AAE_{440\_870}$ of partially coated BC generally first decreases with increasing $D_p/D_c$ and then

increases when $D_p/D_c$ is greater than a certain value. An opposite phenomenon is observed for $ABS_{BrC}$. For BC with $D_f$ of

1.8, $ABS_{BrC}$ of partially coated BC generally increases first with increasing $D_p/D_c$ and then decreases when $D_p/D_c$ is greater

than 2.71.

      With atmospheric aging, the BC cores are reconstructed to be a more compact structure. We used a larger $D_f$ ($D_f$ =2.6) to

represent the compact BC. Even with F = 0, a $D_f$ of 2.6 represents the highly aged BC. By comparing BC with fluffy and

compact structrues, we can see more deeply from the effects of atmospheric aging on the estimations of BrC absorption. As the

BC cores are reconstructed to a compact structure, the AAE = 1 method provides inaccurate estimations even when F = 0 and

$D_p/D_c < 2.15$. As shown in Figure 2 and Table 1, with a $D_f$ of 2.6 and an F of 0, $ABS_{BrC}$ varies in the range of approximately

-13.5% – 13.1% and -5.8% – 20.9% when using $AAE_{440\_675} = 1$ and $AAE_{440\_870} = 1$, repectively. Besides, larger $ABS_{BrC}$

deviations from 0 can be observed as F increases, and $ABS_{BrC}$ sometimes can increases to approximately 20.6% when using

$AAE_{440\_675} = 1$ and to approximately 38.7% when using $AAE_{440\_870} = 1$. Different from the cases where $D_f = 1.8$, BC with a

$D_f$ of 2.6 in most cases exhibits an $ABS_{BrC}$ of larger than 0. The reason is that compact BC generally exhibits a small AAE,

which is generally less than 1 (Luo et al., 2023; Liu et al., 2018). Besides, in most cases, $ABS_{BrC}$ of BC with compact structure

increases with increasing $D_p/D_c$. This can be explained by the findings in the previous study. Luo et al. (2023) have shown

that the $AAE_{440\_870}$ generally decreases with increasing $D_p/D_c$, which leads to an $ABS_{BrC}$ increase with $D_p/D_c$.

      Because the AAE = 1 method would provide inaccurate estimates, many researchers have attempted to use a different AAE

value to estimate BrC. For example, Zhang et al. (2019) used a BC AAE of 0.7 in the Pearl River Delta region, China; Rathod

and Sahu (2022) suggested using a BC AAE of $1.1 \pm 0.05$. However, BC's AAE is subject to large uncertainties in regions, age

status, burn sources, etc. Moreover, the BC AAE itself is not unchanged in the same region, and it would vary with time as the

microphysical properties of BC vary with atmospheric aging. Figures 3 - 4 show how the microscopic properties of BC affect

the application of the different values. As can be seen from Figures 3 - 4, there is no fixed AAE value that is applicable to all

cases. In general, $ABS_{BrC}$ are larger for larger AAE values. Kirchstetter et al. (2004) has measured an AAE range of 0.6 to 1.3.

However, using values in this range, the AAE method would not always provide accurate estimates for BrC in the absence of

additional information. Sometimes the estimated $ABS_{BrC}$ can be approximately -30% – 55% if we choose a fixed AAE in the

range of 0.6 – 1.3. The $ABS_{BrC}$ based on different AAE values is significantly affected by BC size distributions, morphology,

and mixing states. For freshly emitted BC ($D_f = 1.8$, F= 0 in our cases), AAEs of 0.9 – 1.1 would provide reasonable estimates,

and the $ABS_{BrC}$ at this time is in the range of about -6% – 6%. However, using AAEs of 0.6 and 1.3 can sometimes result in

$ABS_{BrC}$ of about -25% and 18%, respectively.

      However, with a more compact structure and a larger F, the choice of accurate AAE values seems to be more important for

the estimation of BrC absorption. As the BC cores become more compact ($D_f = 2.6$), $ABS_{BrC}$ is more affected by the choice

of AAE values. This means that an inaccurate choice of AAE could lead to a larger deviation of BrC absorption in compact

BC, since $ABS_{BrC}$ of compact BC is more sensitive to the AAE values than in fluffy BC. It is difficult to find a fixed AAE

value for estimating BrC in the absence of additional size information, and using a fixed AAE value could result in an $ABS_{BrC}$ range of approximately -26% to 44% in the cases we selected. In Figure 3 – 4, we see that $ABS_{BrC}$ of fluffy BC increases with $r_g$ when AAE is fixed. This is caused by a decrease in AAE with increasing $r_g$ for fluffy BC (Luo et al., 2023; Zhang et al., 2020b). However, in some cases, the AAE of compact BC may increase with increasing $r_g$ (Luo et al., 2023), which could lead to a decrease in $ABS_{BrC}$ with increasing $r_g$. In summary, $ABS_{BrC}$ is significantly affected by BC size when a fixed BC AAE

is used, and it appears that no fixed BC AAE values are applicable to all cases. Similar conclusions can be drawn for the fully coated BC (see Figure S1).

## 3.2    The effects of BC microphysical properties on the WDA method

Because fixed AAE methods cannot always provide accurate estimates of BrC, Wang et al. (2016) proposed a WDA method to derive BrC absorption. They first calculated the WDA based on Mie theory and then derived the AAE based on the WDA.

They claimed that this method can reduce the effects of BC size and coatings. However, in the study of Wang et al. (2016), the morphology of BC was assumed to be spherical, and the WDA calculated based on Mie theory does not always provide accurate estimates. In fact, the spherical model only represents the highly aged BC. Unaged BC in the atmosphere often exhibits fluffy morphologies, and the coating materials make the morphology coated BC more complex (Pang et al., 2022, 2023; Wang et al., 2017). The WDA calculated based on the Mie theory may not represent BC with complex morphologies. In addition, as

BC morphologies can change with atmospheric aging, the spherical assumption provides an uncomplete understanding on the effects of atmopheric aging on WDA. Figure 5 shows the estimated $ABS_{BrC}$ based on the WDA method. Since some BC cores with partial coating are fully coated while others are not, we calculated $ABS_{BrC}$ based on the WDA using the bare sphere model or the core-shell model. As pointed out by Luo et al. (2021c), the WDA method does not necessarily provide better estimates than the AAE = 1 method. The estimated $ABS_{BrC}$ based on the Mie theory-based WDA can vary in a range from

about -40.8% to 35.7% (see Table 2).

The applicability of the WDA method is also related to the atmospheric aging status. As shown in Figure 5 and Table 2, the estimated $ABS_{BrC}$ based on the WDA method is generally in the range of -14.7% – -9.8% for freshly emitted BC (F = 0, $D_f$ = 1.8, $D_p/D_c$ < 2.15), and the range becomes approximately -17.4% – 12.5% when $D_p/D_c$ increases to 4.64. For freshly emitted BC, the estimated $ABS_{BrC}$ is generally less than 0. This is because the WDA estimated with Mie generally underestimates

the WDA of BC with fluffy morphology. As shown in Figure 6 and Table 3, the WDA estimated with the morphologically realistic BC model is not significantly different from 0 for freshly emitted BC (F = 0, $D_f$ = 1.8, $D_p/D_c$ < 2.15), which is about -0.04 – 0, whereas the WDA estimated with the spherical model varies in a range from about -0.30 to -0.14. The estimated $ABS_{BrC}$ is in the range of about -20.5% – 16.1% and -15% – 35.7% when F increases to 0.1 and 0.3, respectively, and the difference in WDA between the partially coated BC and the spherical BC can sometimes exceed 0.4. The estimated $ABS_{BrC}$

based on the WDA method is generally smaller than 0 when the BC cores become compact ($D_f$ = 2.6). This is due to the fact that the WDA of the partially coated BC is underestimated by Mie theory (see Fig. 6). For a $D_f$ of 2.6, the estimated $ABS_{BrC}$ is generally in the range of -40.8% – 0.1%, and the difference between the WDA of the partially coated BC and the spherical BC can sometimes even exceed 0.6. Thus, since Mie theory does not always provide accurate estimates for the WDA of BC with

realistic morphologies, we should carefully consider the effects of the morphologies of partially coated BC when applying the WDA method.

Figure 7 shows the effects of size distributions on estimated $ABS_{BrC}$ based on the WDA method. $ABS_{BrC}$ shows different trends with $r_g$ for BC with different F and $f_{BC}$. When $D_f = 1.8$ and F = 0, $ABS_{BrC}$ estimated by the WDA method generally decreases with increasing $r_g$ when $f_{BC} >$ is 5%, while the opposite phenomenon is observed when $f_{BC} = 1\%$. As F increases, different $ABS_{BrC}$ trends can be observed. Unlike the case of F = 0, $ABS_{BrC}$ increases with increasing $r_g$ when $f_{BC} = 5\%$. As BC cores are reconstructed into a compact structure ($D_f = 2.6$), $ABS_{BrC}$ decreases with increasing $r_g$ when BC is heavily coated ($f_{BC} = 1\%$ and 5%), while an opposite trend is observed for thinly coated BC. Moreover, $ABS_{BrC}$ decreases with increasing $r_g$ even for thinly coated BC when F is larger, and similar findings are found for fully coated BC (F = 1.0), as shown in Figure S2.

To explain why the above phenomenon occurs, we also calculated the WDA of BC with different $r_g$. Figure 8 shows the variations of $ABS_{BrC}$ with $r_g$ for partially coated BC with different mixed states. The WDA of partially coated BC is very different from that of spherical BC. In general, the WDA estimated with the bare sphere model and the core-shell sphere model show similar trends with $r_g$. When F = 0 and $f_{BC} = 5\%$, the WDA calculated with the morphologically realistic model is comparable to that calculated with the spherical model when $r_g$ is small, leading to $ABS_{BrC}$ of about 0 (see Figure 7). However, as $r_g$ increases, the WDA of the morphologically realistic BC increases, whereas the WDA calculated with the spherical models does not change significantly. Thus, the WDA difference between morphologically realistic BC and spherical models (the WDA of the morphologically realistic BC minus the WDA of the spherical BC) increases with increasing $r_g$, so that $ABS_{BrC}$ decreases with $r_g$. However, the WDA of both the morphologically realistic BC and spherical models does not vary significantly with $r_g$ when F = 0 and $f_{BC} = 20\%$, so the $ABS_{BrC}$ does not vary significantly when $r_g$ is changed. When F increases, a different phenomenon is observed. When F = 0.2 and $f_{BC} = 5\%$, the WDA of morphologically realistic BC is much smaller than that of spherical models, which can lead to a positive $ABS_{BrC}$. Moreover, the WDA of morphologically realistic BC decreases significantly with the increase of $r_g$, while the WDA calculated with the spherical models are not significantly varied, so the WDA difference decreases with increasing $r_g$. Therefore, the $ABS_{BrC}$ can increase with $r_g$.

As BC cores become compact ($D_f = 2.6$), different WDA trends are observed with $r_g$. The WDA of partially coated BC is generally larger than that of spherical BC in the cases where F = 0 and $f_{BC} = 5\%$, so that a negative $ABS_{BrC}$ is observed. Moreover, during this time, the WDA of the morphologically realistic BC increases with increasing $r_g$, whereas the WDA calculated with the spherical models do not vary significantly with $r_g$. As a result, the WDA difference between the partially coated BC model and the spherical models increases with increasing $r_g$, leading to decreasing $ABS_{BrC}$ with $r_g$. Similar results are found when F = 0.2 and $f_{BC} = 5\%$, and when F = 0.2 and $f_{BC} = 20\%$. A different phenomenon was observed when F = 0 and $f_{BC} = 20\%$. At this time, the WDA of partially coated BC is generally larger than that of spherical BC, so $ABS_{BrC}$ has negative values. However, the WDA of partially coated BC decreases with increasing $r_g$, and the WDA difference between the morphologically realistic model and the spherical models becomes smaller. Therefore, $ABS_{BrC}$ increases with $r_g$ and tends to 0 when $r_g$ is large (e.g., $r_g$ =0.1). Similar results can be found for other $\sigma_g$, with minor differences observed (see Figure S3 – S4.).

### 3.3 Effects of microphysical properties of BC on the global estimation of BrC

Recent studies have generated increasing interest in estimating the global distribution of BrC (Zeng et al., 2020a; Wang et al., 2016). In this work, the effects of BC microphysical properties on the global estimation of BrC were also investigated. Figure 9 shows the mean global optical absorption aerosol depth (AAOD) of BC calculated with different configurations. In the figures, case A represents aerosols where $D_f = 1.8$, $F = 0.0$, and $f_{BC} = 5\%$; case B represents aerosols where $D_f = 1.8$, $F = 0.0$ and $f_{BC} = 20\%$; Case C represents aerosols where $D_f = 1.8$, $F = 0.2$ and $f_{BC} = 5\%$; Case D represents aerosols where $D_f = 1.8$, $F = 0.2$ and $f_{BC} = 20\%$; Case E represents aerosols where $D_f = 2.6$, $F = 0.0$ and $f_{BC} = 5\%$; Case F represents aerosols where $D_f = 2.6$, $F = 0.0$, and $f_{BC} = 20\%$; Case G represents aerosols with $D_f = 2.6$, $F = 0.2$, and $f_{BC} = 5\%$; Case H represents aerosols with $D_f = 2.6$, $F = 0.2$, and $f_{BC} = 20\%$.

By comparing the five models, Sand et al. (2021) showed that the BC AAOD is generally in the range of 0.0007 – 0.007. Kinne (2019) reported a larger AAOD of 0.0072. However, their studies assume spherical core-shell structures, and a fully coated model could overestimate the overall AAOD. Based on GEOS-Chem simulations, Kelesidis et al. (2022) have shown that the global mean BC AAOD is about $0.0017 \pm 0.007 \pm$ and $0.003 \pm 0.0016$ when the spherical bare model and the coated model are used, respectively. However, the values increase to about $0.0021 \pm 0.0008$ and $0.0036 \pm 0.0014$ when bare and coated BC agglomerate models are used, respectively. However, in their study, the coated BC aerosols were assumed to be fully coated. As shown in Figure 9 and Table 4, our case studies show a global mean AAOD value of about 0.0016 to 0.0026, which strongly depends on the microphysical properties of BC. Our simulated AAOD is generally in the range of Kelesidis et al. (2022). Our simulated BC AAOD is a little larger than simulations with agglomerate-only models by Kelesidis et al. (2022), but smaller than simulations with fully coated agglomerate models. This is easy to understand since we have mainly considered partially coated BC in this work. It is worth noting that the AAOD in this work is 440 nm, while previous studies generally set it at 550 nm. However, the AAOD does not differ significantly between these two wavelengths. Moreover, the aim of this work is to evaluate the impact of BC microphysical properties on the estimation of BrC, but not to make a detailed comparison with the previous studies, so our analysis is valid.

In general, the AAODs of BC with fluffy structure are higher than those of BC with compact structure, which is consistent with the results of previous studies (Liu and Mishchenko, 2005; Luo et al., 2022; Kahnert and Devasthale, 2011). This is due to the blocking effects of a more compact structure and results in lower absorption (Kahnert and Devasthale, 2011). When F = 0, the effects of the coating ratio ($f_{BC}$) are not significant. The global BC AAOD is about $1.9 - 2.1 \times 10^{-3}$ and $1.7 - 1.9 \times 10^{-3}$ for BC with $D_f = 1.8$ and $D_f = 2.6$, respectively. As expected, the global mean BC AAOD increases as F increases. As F increases to 0.2 when $D_f = 1.8$, the global BC AAOD increases to about $2.1 - 2.5 \times 10^{-3}$ and $2.45 - 2.6 \times 10^{-3}$ when $f_{BC}$ is 5% and 20%, respectively. The results also show that the AAOD does not necessarily increase when the coating ratio (decreasing $f_{BC}$) for the partially coated BC increases due to the shielding effect of the coating materials.

BC DRF is an important parameter for assessing climate change. Bond et al. (2013) estimated a global mean BC DRF of about $+0.17 - +1.48$ W/m$^2$, while more recent studies have shown much lower BC DRF values. For example, Kinne (2019) showed a global mean BC DRF of about $+0.55$ W/m$^2$; Matsui et al. (2018) reported a BC DRF of $+0.18 - +0.42$ W/m$^2$;

Chen et al. (2022) estimated a mean BC DRF of +0.33 W/m$^2$ [+0.17, +0.54]. However, Heald et al. (2014) reported a much smaller global mean BC DRF of +0.078 W/m$^2$; Tuccella et al. (2020) showed a global mean BC DRF of about +0.13 and

405 +0.25 W/m$^2$ using a bare-sphere model and a core-shell-sphere model, respectively. As shown in above, we estimate a global mean AAOD range of about 0.0016 to 0.0026. Considering an average absorption force of $170 \pm 43$ W/m$^2$ /AAOD, based on the study of Bond et al. (2013), the estimated DRF in this work may be about $+0.272 \pm 0.069$ to $+0.442 \pm 0.112$ W/m$^2$, which is generally in the range of values reported by previous studies.

Figure 10 and Table 5 show the estimated global BC AAOD incorrectly attributed to BrC (BrC AAOD error in the figures)

when AAE-based methods are used. Using the $AAE_{440\_675} = 1$ method, the errors for cases A, B, E, G, are relatively small, and the AAOD errors are in the range of $-0.1 - 0.08 \times 10^{-3}$. However, in cases C, D, F, and H, the errors are larger, and the estimated AAOD errors can sometimes reach about $0.41 \times 10^{-3}$, which corresponds to about 20% of the global BC AAOD. A wider range for AAOD errors is found in our cases when the method $AAE_{440\_870} = 1$ is used. A general AAOD error range of about $-0.43 - 0.46 \times 10^{-3}$ was estimated when using $AAE_{440\_870} = 1$. Assuming an average absorption forcing efficiency

of $170 \pm 43$ W/m$^2$ /AAOD (Bond et al., 2013), we can estimate a DRF range of $-0.073 \pm 0.0185$ to $+0.078 \pm 0.0198$ W/m$^2$ at BC, which is incorrectly attributed to BrC. The WDA does not necessarily improve the estimates when BC has a complex morphology. The spherical model-based WDA methods may result in a global mean AAOD error range of approximately $-0.87 - 0.04 \times 10^{-3}$ in our cases, which could result in a global mean mis-assigned DRF range of $-0.153 \pm 0.0387$ to $+0.0085 \pm 0.0022$ W/m$^2$.

Figures 11 – 12 and Figures S5 – S6 show the global distribution of BC AAOD mis-attributted to BrC using different AAE-based methods. Our results show that the applicability of the different AAE-based methods is significantly limited by the microphysical properties of BC. When all BC are freshly emitted (e.g., F = 0, $f_{BC}$ = 20%), the $AAE_{440\_870} = 1$ method would provide relative reasonable estimates for BrC AAOD estimates, and the mis-attributed AAOD are within $0.8 \times 10^{-3}$. However, the errors are more substantial the older the BC (with more coating, larger F, or more compact structure). Sometimes

$AAE_{440\_870} = 1$ leads to an AAOD error of about $-7.3 - 5.7 \times 10^{-3}$ in East Asia, resulting in a BC DRF of about $-1.24 \pm 0.314$ W/m$^2$ to $+0.97 \pm 0.245$ W/m$^2$, which is incorrectly attributed to BrC. Similar results are also observed using the $AAE_{440\_675} = 1$ method (see Figure S5.). In contrast, the $AAE_{440\_675} = 1$ method would actually lead to relatively smaller errors for freshly emitted BC. As shown in Figure 12 and Figure S6, the WDA method may lead to larger errors if all BC have complex morphology. In some cases, the Mie theory-based WDA methods would lead to an mis-attributed AAOD of about $-22$

$\times 10^{-3}$ in some regions (e.g., East Asia). Multiplying this AAOD error by an average absorption forcing of $170 \pm 43$ W/m$^2$ /AAOD results in an estimated mis-assigned DRF of $-3.74 \pm 0.946$ W/m$^2$. Therefore, we should carefully consider the effects of BC microphysical properties when using AAE-based methods.

## 4 Atmospheric implication

AAE-based methods have been widely used to estimate the absorption of BrC, while they are subject to large uncertainties

due to the properties of BC. We quantify the effects of the microphysical properties of BC based on numerical simulations and

investigate how the applicability of AAE-based methods varies at different aging conditions. From the above, it is clear that using a BC AAE of 1 can provide reasonable estimates for BrC absorption, while the deviation from the "true" BrC absorption becomes significant as the particles age. This means that the AAE = 1 method can provide inaccurate estimates when aged BC is present. In general, regions near emission sources, such as urban traffic areas, contain mainly freshly emitted BC. In this case, it is reasonable to use the AAE = 1 method. With atmospheric aging, we should adjust the AAE values because both the AAE = 1 and WDA methods can sometimes result in misallocations of tens of percent of BrC absorption. However, the adjustments should differ depending on the aging condition. As shown in Figure 3 – 4, for fluffy BC partially mixed with coating materials ($D_f$ = 1.8 and 0 < F < 1 in this work), $ABS_{BrC}$ = 0 occurs in most cases when AAE > 1. Therefore, we generally propose a relatively larger AAE, while a smaller AAE is recommended for compact BCs, including coated and uncoated compact BCs. Recent observations have shown that the average $D_f$ is often small even for coated BCs in regions far from emission sources (Wang et al., 2017; Yuan et al., 2019). Therefore, we prefer larger AAEs. However, there are also some compact BC aerosols in the atmosphere, and we should also consider the uncertainties when the real BC aerosols have a compact structure. In addition, the WDA methods does not improve the estimation. Therefore, we should carefully consider the uncertanties caused by the microphysical properties of BC when estimating the BrC absorption and DRF based on the AAE based methods.

## 5    Summary and conclusions

The AAE-based method is commonly used to estimate the absorption of BrC, but may provide inaccurate estimates due to the effects of the microphysical properties of BC. The goal of this work is not to discuss the use of the AAE-based method, but to assess the uncertainties of the AAE-based method. We find that an AAE of 1 can provide a reasonable estimate when BC is freshly emitted. Therefore, an AAE of 1 is suggested for regions close to the emission source, such as vehicle emission region. However, we should also note the uncertainties associated with using an AAE of 1. We estimate an $ABS_{BrC}$ range of about -4.8% to 2.7% when using an AAE of 1 for freshly emitted BC. However, the $ABS_{BrC}$ range becomes broader when BC is aged, and sometimes $ABS_{BrC}$ can be varied in the range of about -34.5% – 38.7%, depending on the aging status and morphologies. Therefore, we need to adjust the AAE value when the fixed AAE method is applicable to the region consisting of aged BC, such as regions far from the emission source. However, even for aged BC, different AAE values should be used for different aging conditions, since we show that no fixed AAE is applicable for all cases.

This work represents the aging condition by assuming a more compact structure, more coating materials and a larger F. For different aging processes, the adjustment of AAE values should be different. For fluffy BC partially mixed with coating materials ($D_f$ = 1.8 and 0<F<1 in this work), we generally propose a larger AAE, while a smaller AAE is recommended for the compact BC. Our results also show that the Mie theory-based WDA method does not necessarily improve the estimate, with a corresponding $ABS_{BrC}$ range of about -40.8% – 35.7% in our simulation cases, due to the substantial WDA deviation between the morphologically realistic BC and the spherical BC.

At the global level, the use of BC AAE of 1 can lead to a global mean mis-assigned AAOD of about $-0.43 – 0.46 \times 10^{-3}$, resulting in a corresponding global mean mis-assigned DRF of $-0.073 \pm 0.0185$ to $+0.078 \pm 0.0198$ W/m$^2$. However, for the

freshly emitted BC, an AAE of 1 does not lead to a significant misestimation of the AAOD. At the regional level, for an AAE

of 1, the mean mis-assigned AAOD can vary in the range of -7.3 to $5.7 \times 10^{-3}$ in some regions, leading to a mis-assigned DRF of about $-1.24 \pm 0.314$ W/m$^2$ to $+0.97 \pm 0.245$ W/m$^2$. The WDA method does not necessarily provide more accurate estimate for BrC absorption, and sometimes in some regions we can see a mean mis-assigned AAOD of about $-22 \times 10^{-3}$, leading to a mis-assigned DRF of about $-3.74 \pm 0.946$ W/m$^2$. Therefore, the effects of the microscopic properties of BC should be carefully considered when estimating BrC absorption and its direct radiative forcing based on the measurements at

multiple wavelengths.

*Acknowledgements.* We gratefully acknowledge financial support from the National Key R&D Program of China (Grant No.2022YFB3902802) and the National Natural Science Foundation of China (Grant No. 42305148 and 41871269). We particularly thank Dr. Michael Mischenko for making the T-matrix code publicly available.

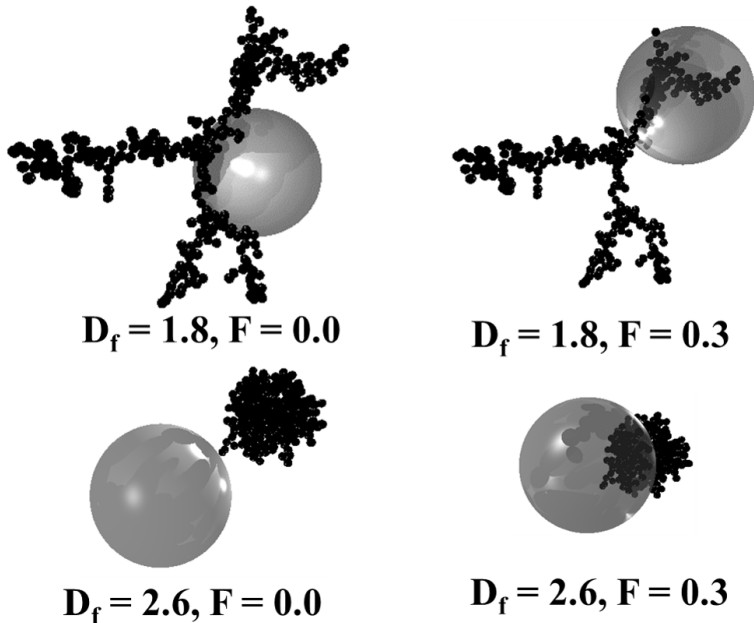

**Figure 1.** Typical BC morphologies assumed in this work, similar to Luo et al. (2023). We used a $D_f$ of 1.8 and 2.6 to represent fluffy BC and compact BC, respectively. Moreover, F = 0 means that BC is not internally mixed with coating materials, while BC is gradually partially coated with other materials as F increases. F = 1 means that BC is completely coated.

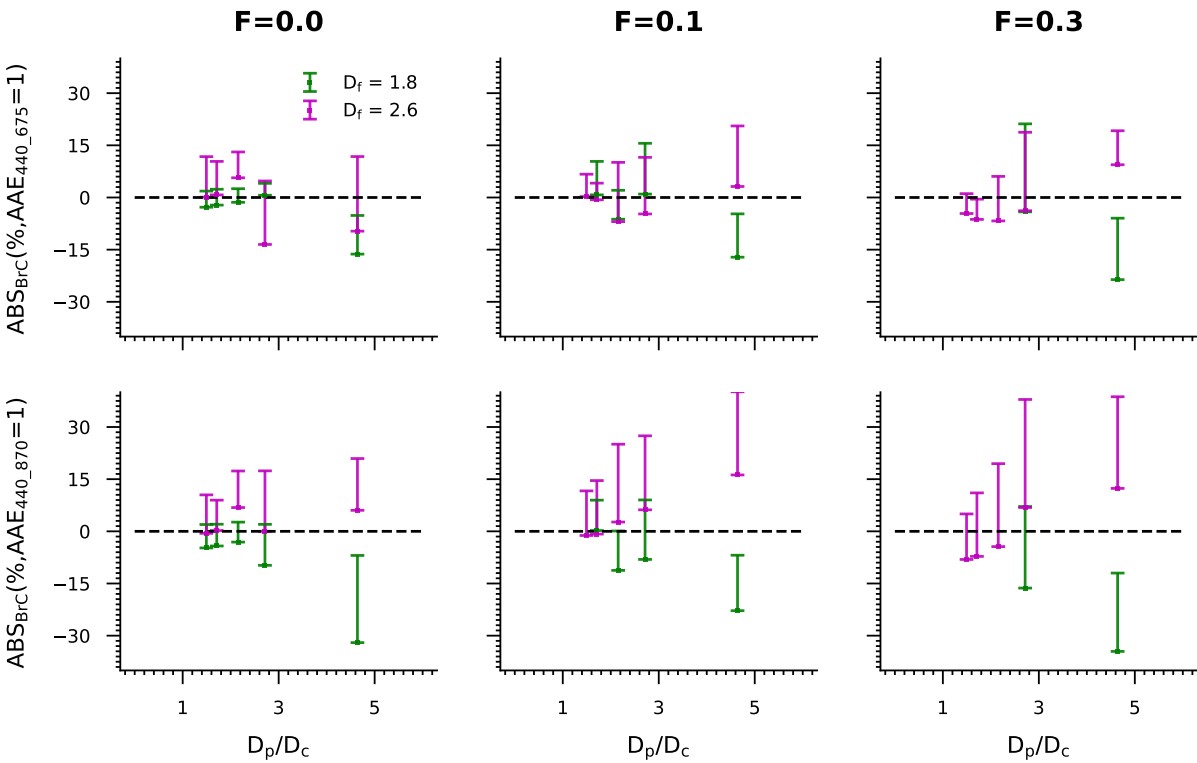

**Figure 2.** $ABS_{BrC}$ at different $D_p/D_c$ estimated using the AAE = 1 methods. The error bars in the figures represent the upper and lower limits when $r_g$ is varied in the range of $0.05 - 0.1$ $\mu m$ and $\sigma_g$ is varied in the range of $1.5 - 1.8$. When freshly emitted, BC generally exhibits a fluffy structure and is not internally mixed with coating materials, as reflected by an F of 0 and a $D_f$ of 1.8. However, with increasing atmospheric aging, BC gradually becomes internally mixed with coating materials and thicker coating materials overlay on the BC, which may be reflected in a larger F and $D_p/D_c$. Moreover, the BC structure becomes more compact as the particles age, and we used a large $D_f$ value to represent the compact BC.

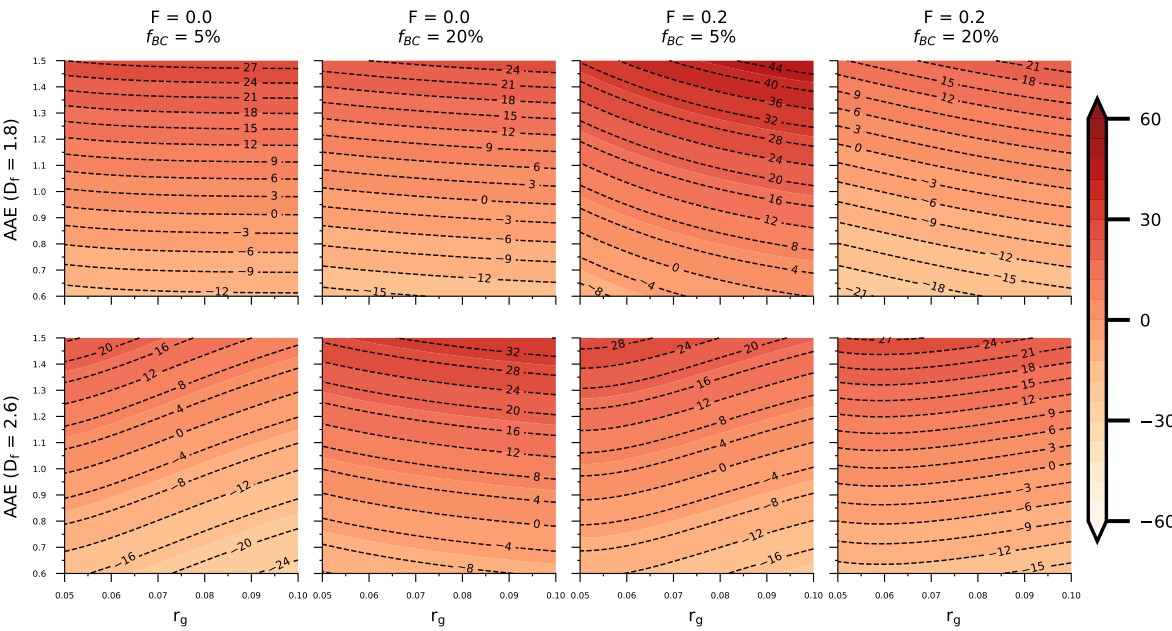

**Figure 3.** The variations of $ABS_{BrC}$ (%) estimated based on the fixed AAE with the function of AAE and $r_g$, where the wavelength pair is 440 nm - 675 nm. We see that $ABS_{BrC}$ increases with $r_g$ when AAE is fixed for fluffy BC, which caused by a decrease in AAE with increasing rg for fluffy BC. However, sometimes opposite phenomenon can be found for the compact BC,

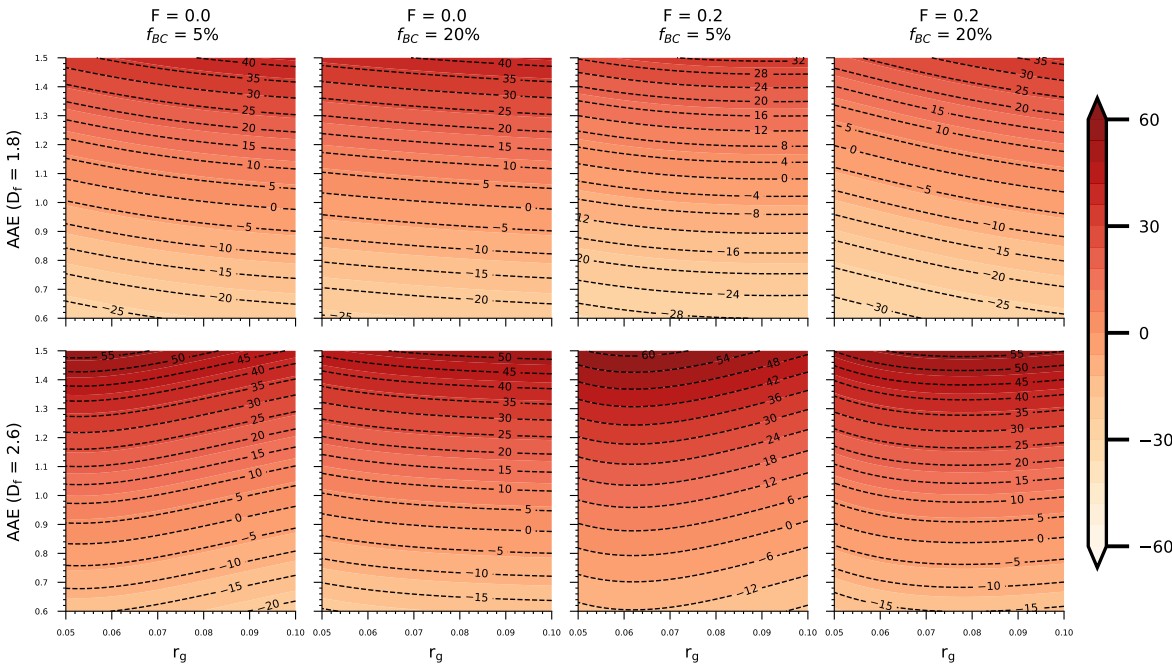

**Figure 4.** Similar to Figure 3, but the wavelength pair is 440 – 870 nm.

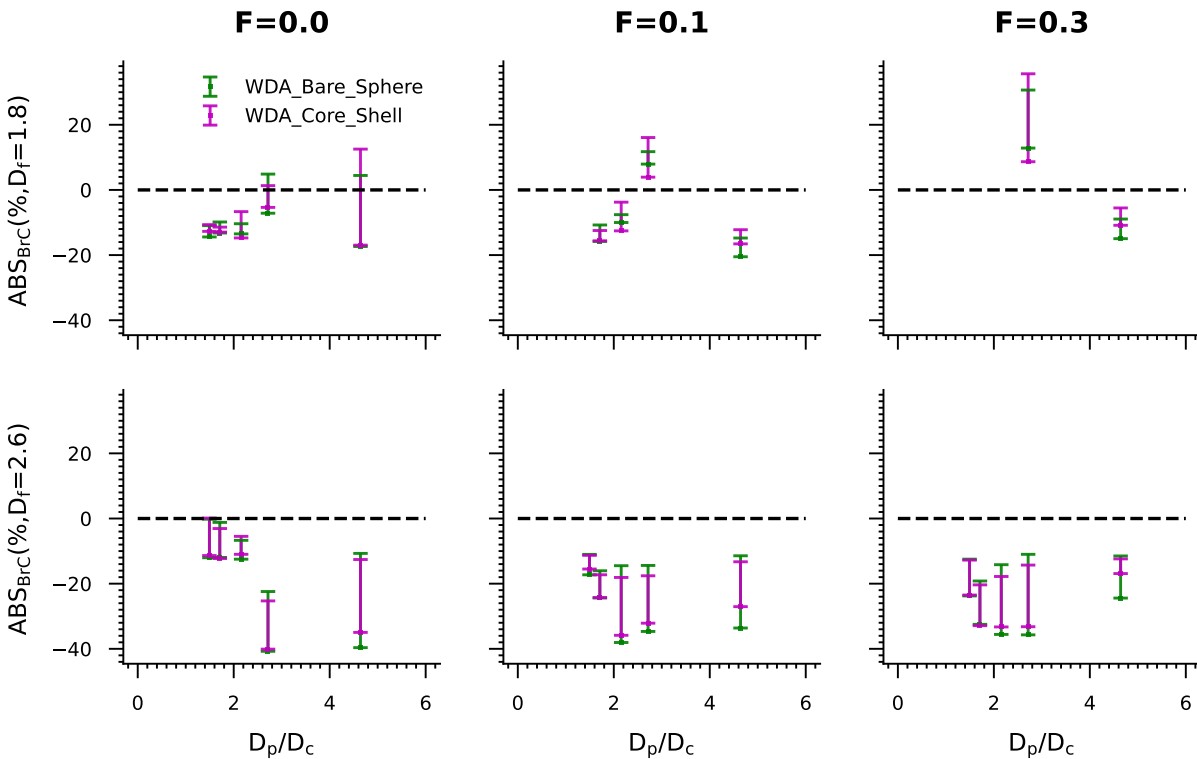

**Figure 5.** Similar to Figure 2, but using the WDA method.We see that the WDA method does not always provide better estimates than the fixed AAE methods, and its applicability is also significantly affected by morphology and mixing states.

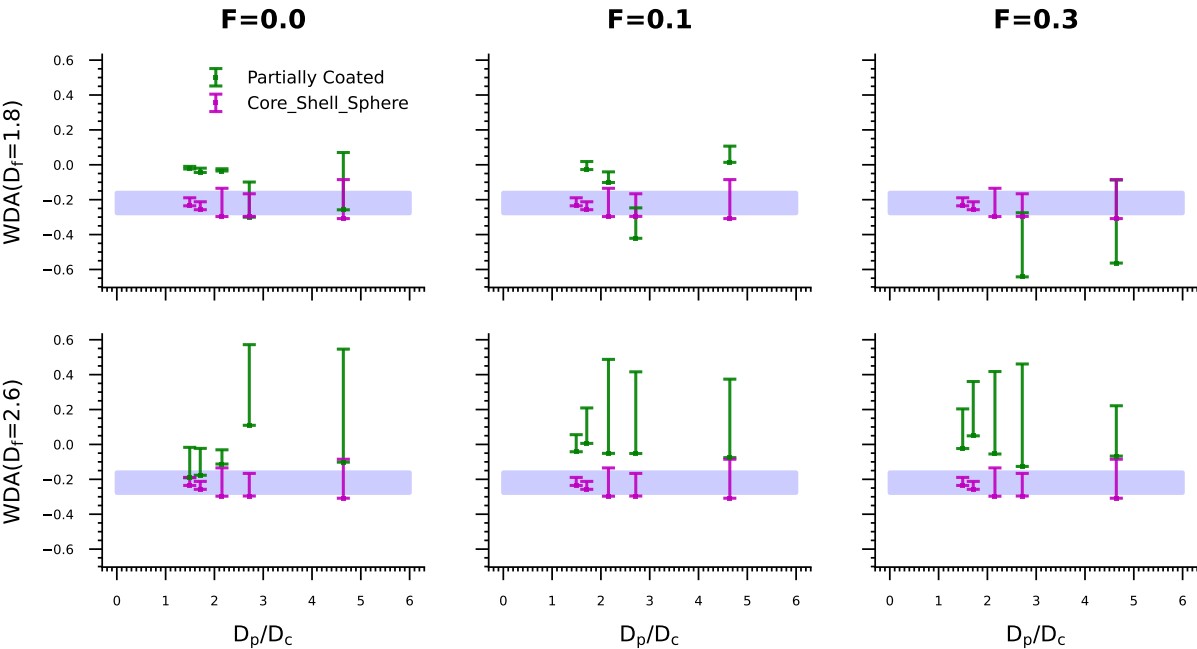

**Figure 6.** The WDA of BC with different morphologies at different $D_p/D_c$, where the purple shading represents the WDA calculated for the bare spherical BC model. The results show that the WDA of BC is significantly affected by morphology and mixture state. When BC is freshly emitted (represented by a $D_f$ of 1.8, an F of 0, and a smaller $D_p/D_c$), the WDA is close to 0. With atmospheric aging, the WDA range becomes broader and often deviates significantly from 0. In addition, the WDA calculated using the Mie assumption can differ greatly from the WDA of partially coated BC, which is why the $ABS_{BrC}$ estimated using the WDA method differs from 0.

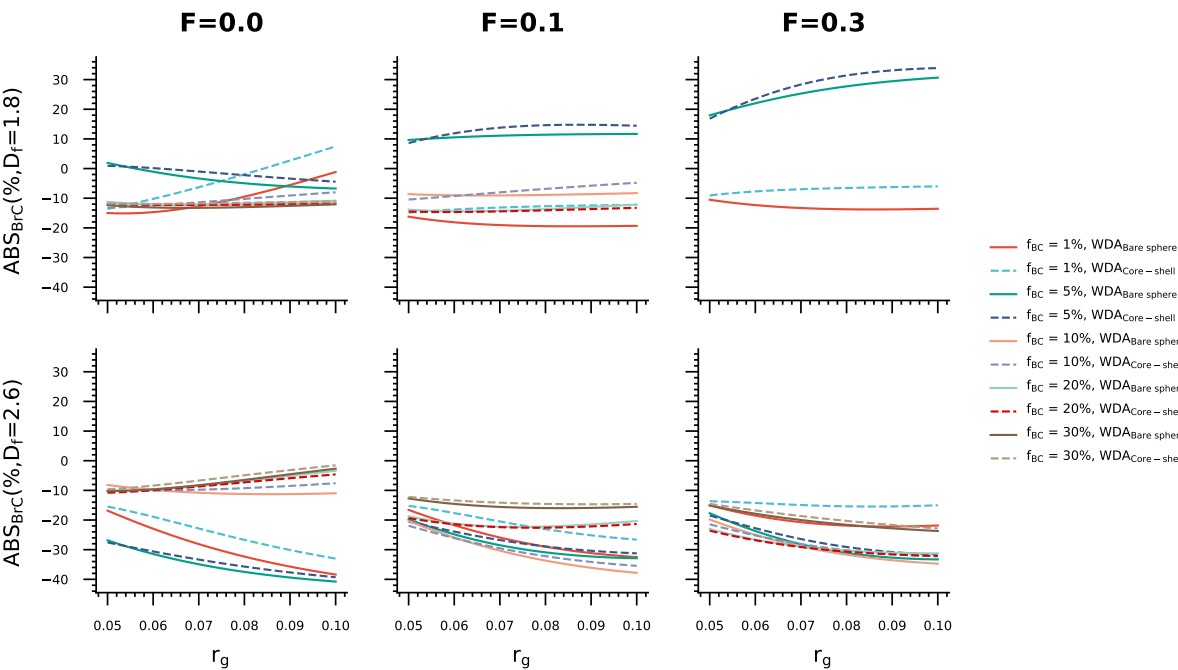

**Figure 7.** The variations $ABS_{BrC}$ estimated using the WDA method with $r_g$ at different mixing states.

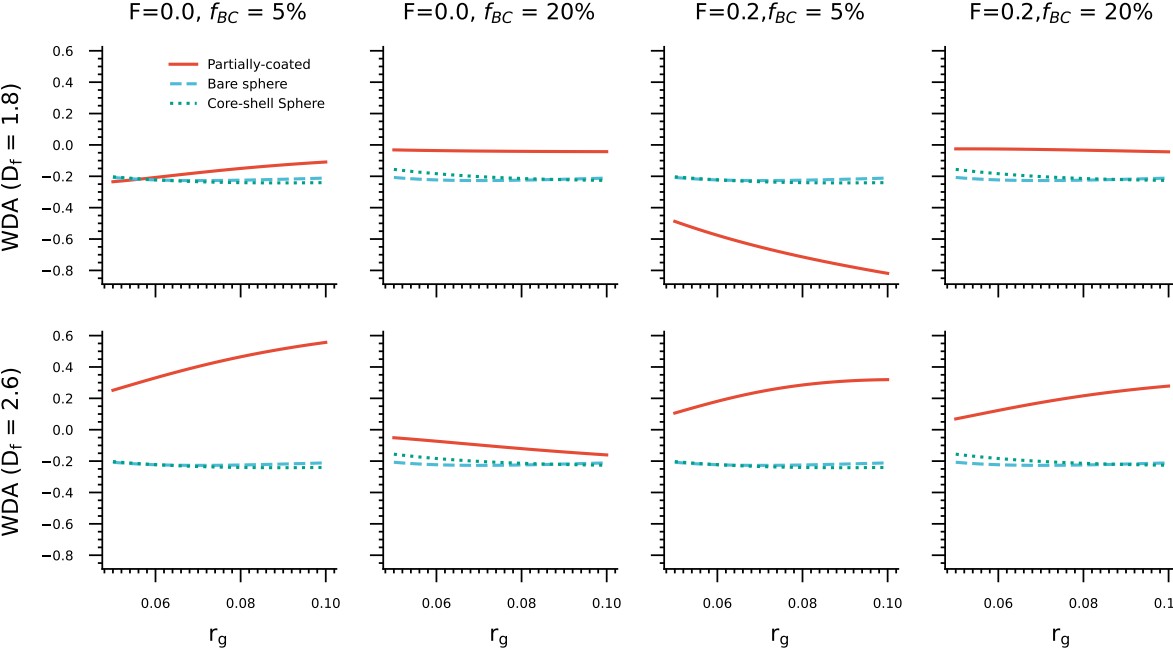

**Figure 8.** The variations WDA of BC with different morphologies with $r_g$ at different mixing states, where $\sigma_g = 1.6$. We can see different variations with $\sigma_g$ for partially-coated BC models and spherical models.

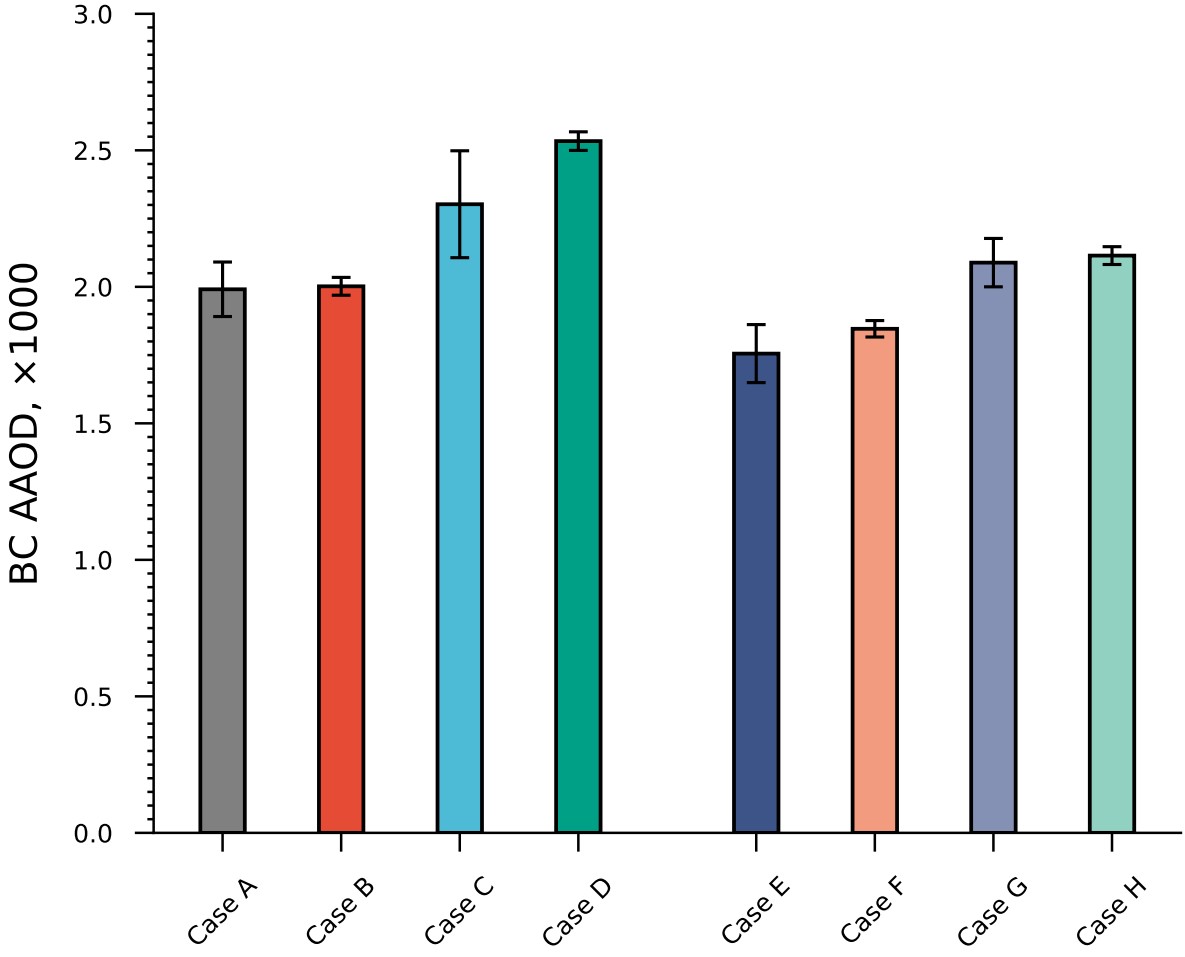

**Figure 9.** The global mean BC AAOD was calculated using different models. The error bars in the figures represent the upper and lower limits when $r_g$ is varied in the range of $0.05 - 0.1$ $\mu$m and $\sigma_g$ is varied in the range of $1.5 - 1.8$. Our case studies show a global mean AAOD value of about 0.0016 to 0.0026, which is significantly affected by mixing states and morphology. The AAOD of BC with a fluffy morphology is generally larger than that of BC with a compact morphology. Moreover, the predicted AAOD does not necessarily increase with increasing coating ratios (decreasing $f_{BC}$), which is due to the shielding effects of the coating materials. Moreover, the AAOD of BC with thicker coating materials (smaller $f_{BC}$) is more sensitive to the size distribution.

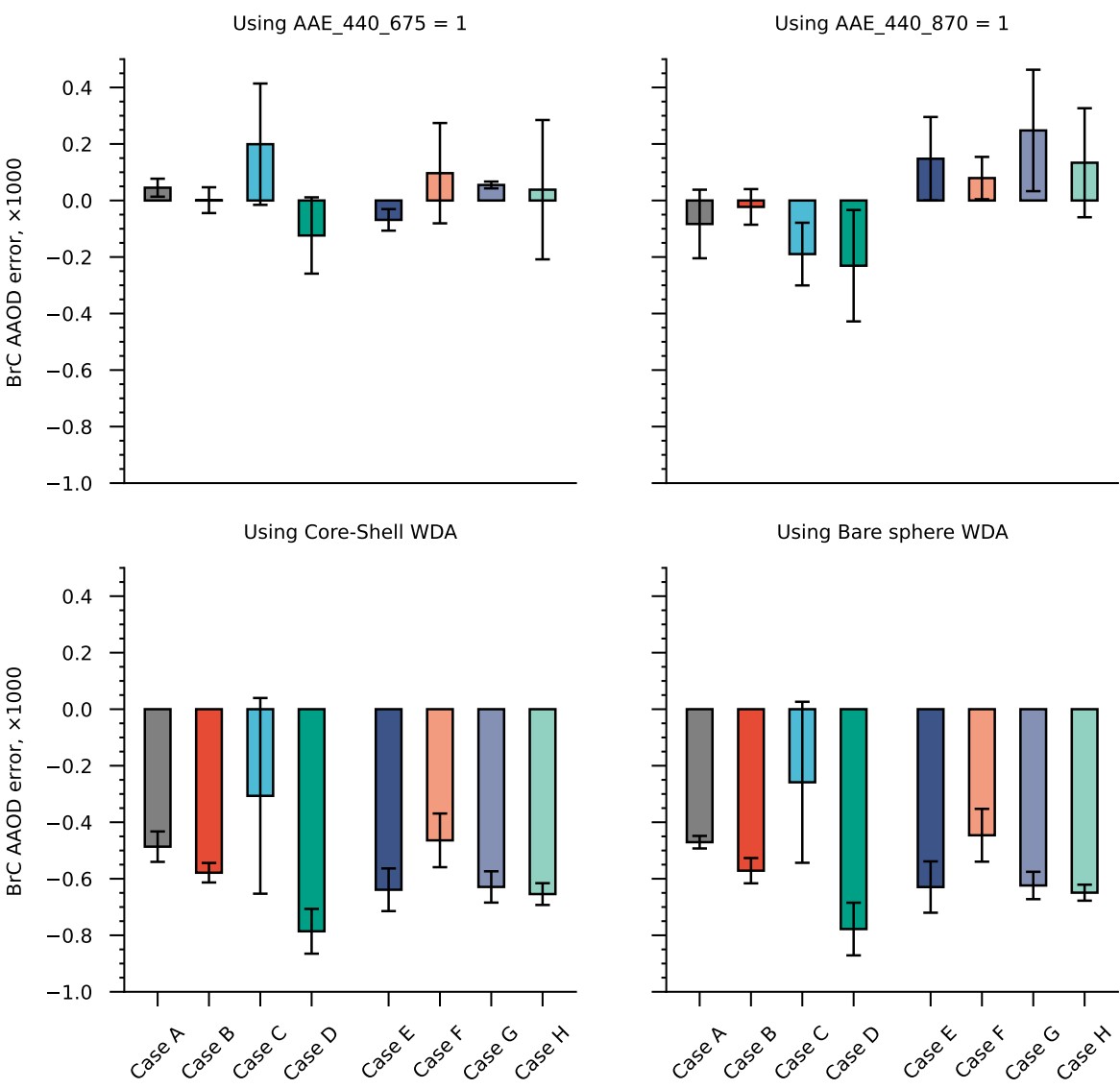

**Figure 10.** The global mean BC AAOD that is misattributed to BrC for different mophological configurations. The error bars in the figures represent the upper and lower limits when $r_g$ is varied in the range of $0.05 - 0.1$ $\mu$m and $\sigma_g$ is varied in the range of $1.5 - 1.8$.

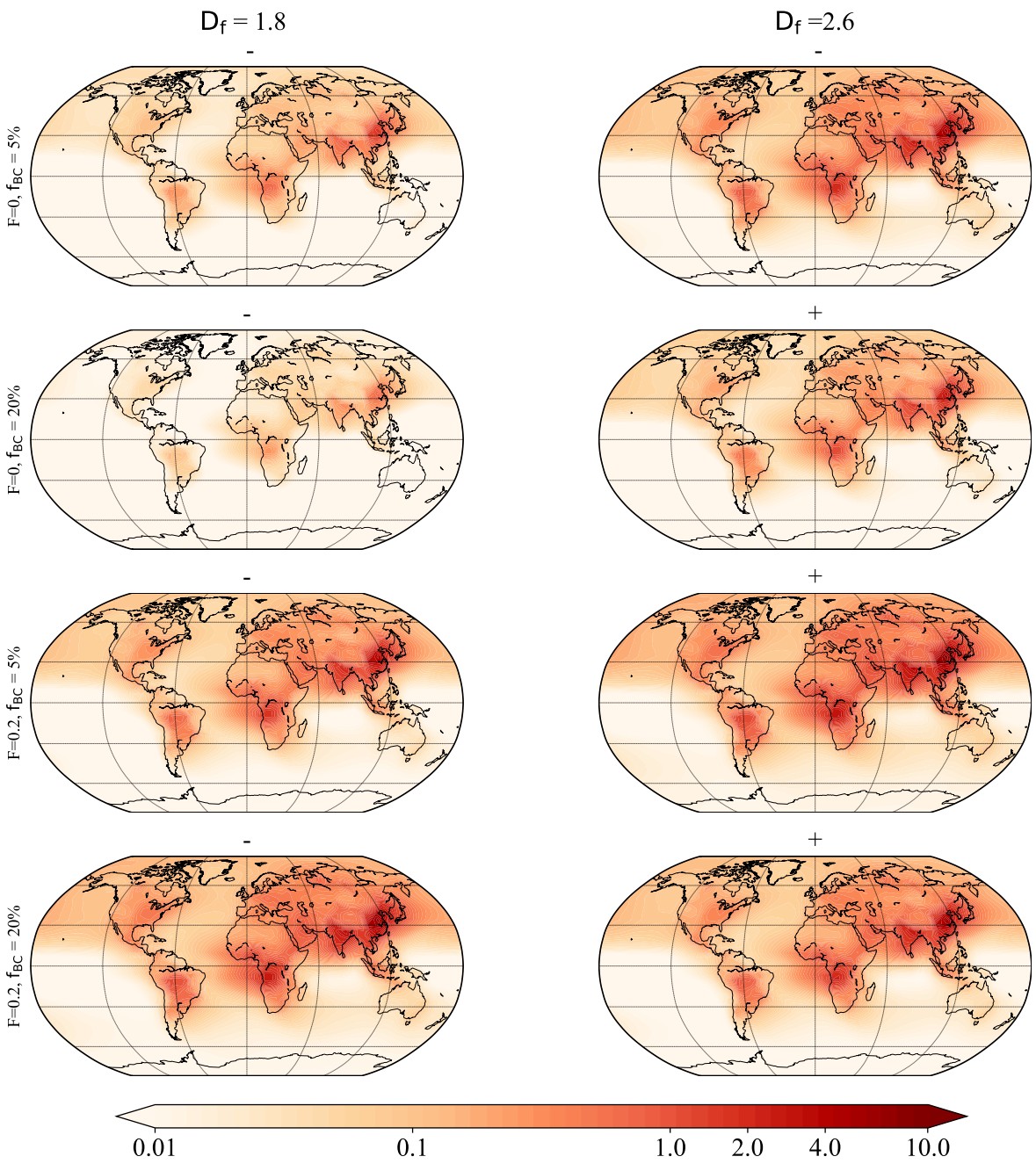

**Figure 11.** The global distributions of BC AAOD ($\times1000$) that is miattributed BrC based on the $AAE_{440\_870} = 1$ method, where negative sign means underestimation, and positive sign means overestimation.

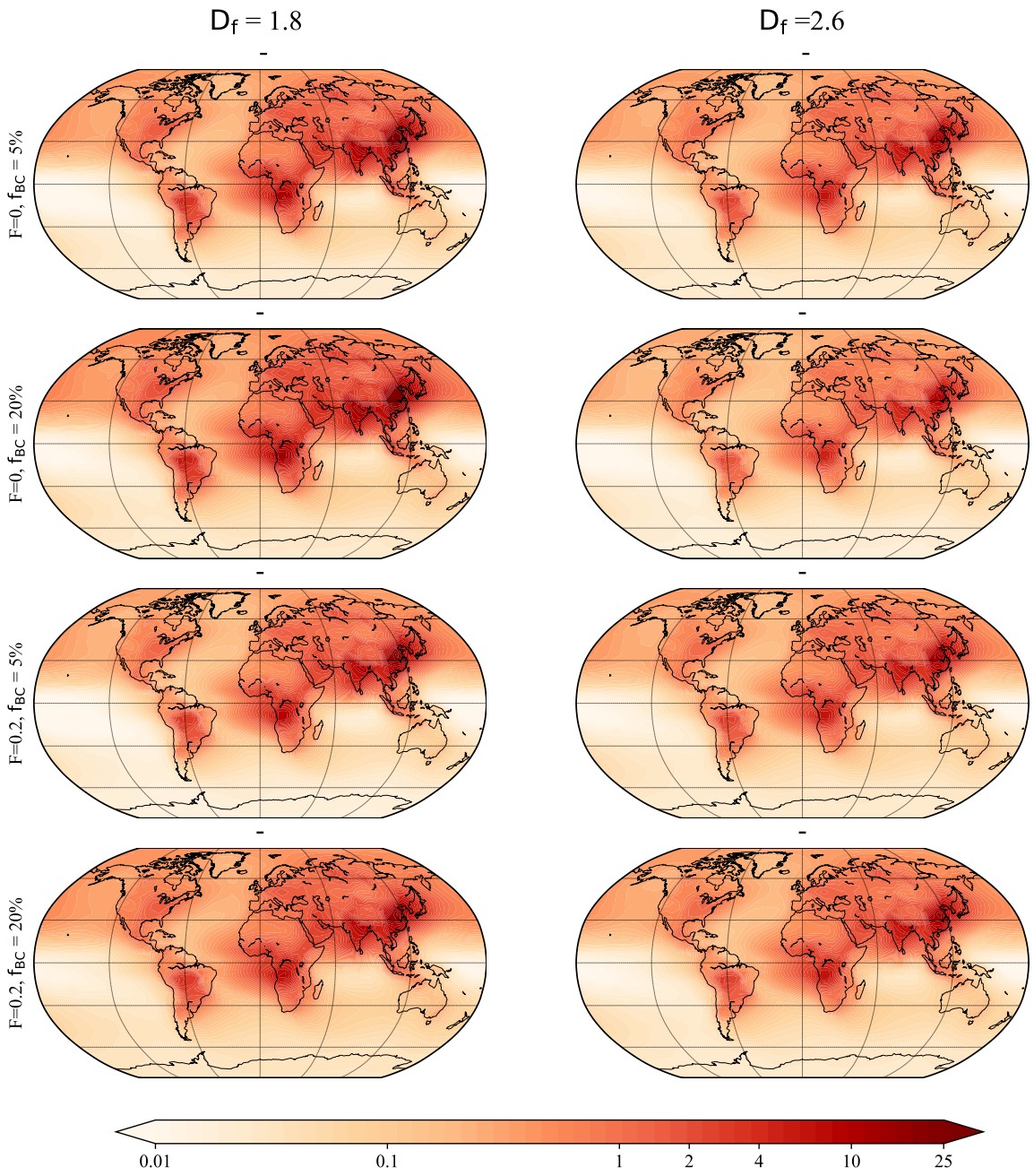

**Figure 12.** Similar to Figure 11, but using the bare sphere WDA method.

**Table 1.** $ABS_{BrC}$ (%) at different $D_p/D_c$ estimated using the AAE = 1 methods, as shown in Figure 2.

| Method | F | $D_f$ | $D_p/D_c$= 1.49 | $D_p/D_c$=1.71 | $D_p/D_c$=2.15 | $D_p/D_c$=2.71 | $D_p/D_c$=4.64 |
|---|---|---|---|---|---|---|---|
| Using $AAE_{440\_675}$ = 1 | F = 0 | $D_f$=1.8 | -2.81 – 1.84 | -2.19 – 2.38 | -1.39 – 2.52 | 0.64 – 4.07 | -16.28 – -5.17 |
| | F = 0.1 | $D_f$=1.8 | – | 0.75 – 10.38 | -6.25 – 2.07 | 0.98 – 15.58 | -17.17 – -4.71 |
| | F = 0.3 | $D_f$=1.8 | – | – | – | -4.12 – 21.18 | -23.61 – -5.95 |
| | F = 0 | $D_f$=2.6 | 0.10 – 11.75 | 0.75 –10.38 | 5.66 – 13.09 | -13.51 – 4.75 | -9.68 – 11.77 |
| | F = 0.1 | $D_f$=2.6 | 0.26 – 6.70 | -0.65 – 4.13 | -6.95 – 10.10 | -4.73 – 11.54 | 3.16 – 20.57 |
| | F = 0.3 | $D_f$=2.6 | -4.61 – 1.10 | -6.30 – -0.48 | -6.74 – 6.07 | -3.81 – 18.75 | 9.42 – 19.19 |
| Using $AAE_{440\_870}$ = 1 | F = 0 | $D_f$=1.8 | -4.78 – 1.96 | -4.21 – 2.05 | -3.09 – 2.65 | -9.78 – 2.00 | -32.00 – -6.93 |
| | F = 0.1 | $D_f$=1.8 | – | 0.22– 8.96 | -11.23 – 0.10 | -8.08 – 9.03 | -22.82 – -6.87 |
| | F = 0.3 | $D_f$=1.8 | – | – | – | -16.32 – 7.13 | -34.53 – -12.00 |
| | F = 0 | $D_f$=2.6 | -5.76–10.48 | 0.22 –8.96 | 6.84 – 17.35 | 0.01 – 17.39 | 6.04 – 20.93 |
| | F = 0.1 | $D_f$=2.6 | -1.19–11.63 | -0.79–14.59 | 3.02–25.46 | 6.18–27.46 | 16.22 – -6.87 |
| | F = 0.3 | $D_f$=2.6 | -8.08 – 5.00 | -7.21 – 11.05 | -4.39 – 19.46 | 6.85 – 37.92 | 12.34 – 38.70 |

**Table 2.** $ABS_{BrC}$ (%) at different $D_p/D_c$ estimated using the WDA methods, as shown in Figure 5.

| Method | F | $D_f$ | $D_p/D_c$= 1.49 | $D_p/D_c$=1.71 | $D_p/D_c$=2.15 | $D_p/D_c$=2.71 | $D_p/D_c$=4.64 |
|---|---|---|---|---|---|---|---|
| WDA bare Mie | F = 0 | $D_f$=1.8 | -14.44 – -10.96 | -13.22 – -9.84 | -13.49 – -10.38 | -7.15 – 4.86 | -17.36 – 4.44 |
| | F = 0.1 | $D_f$=1.8 | – | -15.81 – -10.76 | -10.00 – -7.59 | 7.92 – 11.75 | -20.51 – -14.76 |
| | F = 0.3 | $D_f$=1.8 | – | – | – | 12.85 – 30.66 | -14.96 – -8.94 |
| | F = 0 | $D_f$=2.6 | -12.05 – -0.21 | -11.95 – -1.19 | -12.48 – -6.70 | -40.76 – -22.41 | -39.64 – -10.71 |
| | F = 0.1 | $D_f$=2.6 | -17.28 – -11.01 | -24.39 – -16.04 | -38.06 – -14.48 | -34.67 – -14.41 | -33.62 – -11.44 |
| | F = 0.3 | $D_f$=2.6 | -23.76 – -12.48 | -32.49 – -19.18 | -35.58 – -14.17 | -35.71 – -10.99 | -24.42 – -11.49 |
| WDA core-shell Mie | F = 0 | $D_f$=1.8 | -12.72 – -10.65 | -12.94 – -11.42 | -14.74 – -6.65 | -5.36 – 1.32 | -16.95 – 12.53 |
| | F = 0.1 | $D_f$=1.8 | – | -15.61– -12.46 | -12.57 – -3.76 | 3.92 – 16.08 | -16.56 – -12.21 |
| | F = 0.3 | $D_f$=1.8 | – | – | – | 8.66 – 35.65 | -10.86 – -5.53 |
| | F = 0 | $D_f$=2.6 | -11.37 – 0.14 | -12.27 – -3.08 | -10.99 – -5.48 | -40.09 – -25.29 | -34.96 – -12.59 |
| | F = 0.1 | $D_f$=2.6 | -11.50– -11.30 | -24.26 – -17.27 | -35.85– -18.09 | -32.13– -17.58 | -27.06 – -13.28 |
| | F = 0.3 | $D_f$=2.6 | -23.49 – -12.76 | -32.89 – -20.37 | -33.28 – -17.80 | -33.22 – -14.29 | -16.90 – -12.39 |

**Table 3.** The WDA of BC with different morphologies at different $D_p/D_c$ given a typical size distibution range, as shown in Figure 6. The WDA calculated using the bare sphere model is about -0.277 – -0.161.

| Model | F | $D_f$ | $D_p/D_c$= 1.49 | $D_p/D_c$=1.71 | $D_p/D_c$=2.15 | $D_p/D_c$=2.71 | $D_p/D_c$=4.64 |
|---|---|---|---|---|---|---|---|
| Partially coated | F= 0 | $D_f$ = 1.8 | -0.023 – -0.009 | -0.044 – 0.019 | -0.036 – -0.023 | -0.301 – -0.099 | -0.258 – 0.070 |
| Partially coated | F= 0.1 | $D_f$ = 1.8 | – | -0.027 – 0.019 | -0.102 – -0.041 | -0.421 – -0.247 | 0.014 – 0.107 |
| Partially coated | F= 0.3 | $D_f$ = 1.8 | – | – | – | -0.642 – -0.275 | -0.563 – -0.087 |
| Partially coated | F= 0 | $D_f$ = 2.6 | -0.191 – -0.017 | -0.177 – -0.022 | -0.112 – -0.031 | 0.110 – 0.572 | -0.102 – 0.546 |
| Partially coated | F= 0.1 | $D_f$ = 2.6 | -0.042 – 0.056 | 0.006 – 0.209 | -0.043 – 0.491 | -0.052 – 0.416 | -0.076 – 0.374 |
| Partially coated | F= 0.3 | $D_f$ = 2.6 | -0.024 – 0.204 | 0.049 – 0.361 | -0.055 – 0.418 | -0.126 – 0.461 | -0.067 – 0.222 |
| Core-Shell | – | – | -0.235 – -0.189 | -0.257 – -0.212 | -0.297– -0.135 | -0.296 – -0.166 | -0.308 – -0.085 |

**Table 4.** The global mean BC AAOD ($\times$1000) calculated using different models, as shown in Figure 9.

| Case A | Case B | Case C | Case D | Case E | Case F | Case G | Case H |
|---|---|---|---|---|---|---|---|
| 1.89 – 2.09 | 1.97 – 2.03 | 2.11 – 2.50 | 2.50 – 2.57 | 1.63 – 1.88 | 1.72 – 1.97 | 2.00 – 2.18 | 1.94 – 2.29 |

**Table 5.** The global mean BC AAOD ($\times$1000) that is misattributed to BrC for different mophological configuritions, as shown in Figure 10.

| Method | Case A | Case B | Case C | Case D | Case E | Case F | Case G | Case H |
|---|---|---|---|---|---|---|---|---|
| Using $AAE_{440\_675}$ = 1 | 0.013 – 0.077 | -0.044 – 0.047 | -0.015 – 0.414 | -0.259 – 0.011 | -0.107 – -0.031 | -0.081 – 0.274 | 0.043 – 0.067 | -0.208 – 0.285 |
| Using $AAE_{440\_870}$ = 1 | -0.204 – 0.038 | -0.086 – 0.040 | -0.300 – -0.079 | -0.428 – -0.034 | 0.0 – 0.295 | 0.004 – 0.154 | 0.033 – 0.463 | -0.059 – 0.327 |
| Using Core-Shell WDA | -0.540 – -0.433 | -0.613 – -0.544 | -0.653 – -0.040 | -0.865 – -0.707 | -0.714 – -0.563 | -0.559 – -0.369 | -0.684 – -0.574 | -0.693 – -0.616 |
| Using Bare Sphere WDA | -0.493 – -0.448 | -0.616 – -0.527 | -0.543 – 0.026 | -0.871 – -0.685 | -0.720 – -0.538 | -0.540 – -0.353 | -0.672 – -0.575 | -0.678 – -0.621 |

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

*Data availability.* The data can be requested from the corresponding athour.

*Author contributions.* JL conceived the presented idea. JL developed the models, performed the computations, and wrote the paper. JQ verified the simulation methods and results. revised the paper and supervised the findings of this work. All authors discussed the results and contributed to the final paper.

*Competing interests.* The authors declare that they have no conflict of interest.