# Peer review of "Quantifying the effects of the microphysical properties of black carbon on the determination of brown carbon using measurements at multiple wavelengths"

_EGUsphere, 2023_

## Author Response (AR1)

**Response to the comments of editors and reviewers**

(The responses are highlighted in blue)

The authors are very grateful to the editors and reviewers for their valuable comments and constructive suggestions. The reviewers' questions and comments are highlighted in **black font**, and the answers in **blue**. The changes made in the revised manuscript are highlighted in **red.**

**Response to the comments of Editors**

**Comments:** Please ensure that the colour schemes used in your maps and charts allow readers with colour vision deficiencies to correctly interpret your findings. Please check your figures using the Coblis – Color Blindness Simulator (https://www.color-blindness.com/coblis-color-blindness-simulator/) and revise the colour schemes accordingly.

**Reply:** Thanks very much for your comments. We have replotted the figure.

**Response to the comments of Reviewer #1**

**Comments:** This study calculated the optical properties and radiative effects of black carbon (BC) particles with different absorption Ångström exponent (AAE) methods, and further estimated their effect on brown carbon (BrC). The manuscript explained how the microphysical properties of BC particles determine wrong BC's and BrC's absorption and radiative properties under two AAE methods. The study is interesting and the results are helpful for understanding why the optical and radiative properties of BrC is deviated to BC. Overall, the manuscript can be revised and then may be published in ACP. The problems are addressed as following:

**Comments:** The author calculated the absorption deviation for BrC by using non-absorbing coating. If these calculations make sense, we must assume that the true AAE of the BC particles with BrC coating is the same with those with non-absorbing coating. However, is that true? And hence I concern whether the "babs_BC_440_Estimated - babs_BC_440" can represent the Δ I'm very confused here.

**Reply:** thank you for your comment. Here we have only calculated the absorption of black carbon with non-absorbing coating materials, and the BrC absorption is the difference between the total absorption and the black carbon mixed with non-absorbing coating materials. We have assumed that the total absorption at the near-infrared wavelength comes entirely from BC mixed with non-absorbing coatings. Thus, if we know the AAE of BC with non-absorbing coatings, we can calculate the absorption of BC with non-absorbing coating materials, including BC absorption and lensing effect, but excluding the absorption of BrC. Of course, the block effect of BrC would also affect the total absorption, but it is also caused by BrC absorption, so we attribute the effect to BrC absorption. Therefore, to calculate the BrC absorption, we need the AAE of BC mixed with non-absorbing materials, but not the AAE of BC with BrC coating materials. If we know the AAE of BC mixed with BrC, we can only calculate the total absorption, but not the absorption of BC with non-absorbing coatings, and therefore we cannot separate the absorption of BC and BrC.

**Comments:** Besides, refractory index is one of the factors determining the absorption. The deviation of BC absorption will be also affected by the refractory index. All the discussion in this manuscript is carried out under the 1.95+0.79i for BC and 1.55+0i for coating. The results for "the estimated BrC absorption should be the absorption from BC that is incorrectly attributed to BrC" is not comprehensive because the refractive index varies.

**Reply:** From the review by Bond and Bergstrom (2006), it appears that the value 1.95+0.79i should be used at 550 wavelengths. While it is true that the BC refractive index can vary with wavelength, it deviates significantly in the UV and NIR. In addition,

it is still unclear how the BC refractive index varies with wavelengths, and we do not expect much variation from the UV and NIR, as shown in Bond and Bergstrom (2006),. Therefore, it is often assumed that the BC refractive index is independent of wavelength in the UV and NIR. Furthermore, if we use a spectral dependent refractive index, we cannot see how the microscopic properties affect the measurements, since the BC refractive index would also affect the total absorption. Therefore, we used a fixed refractive index. It is difficult to do all the calculations because the actual refractive index of BC is still an open question. However, since our study focuses on the effects of microscopic properties, we used a fixed refractive index. Even though we only consider a fixed refractive index, we believe that the main conclusions are similar for other refractive indices.

**Comments:** The study may have an important implication on the estimation of BrC's optical properties and radiative effect. This should be addressed in the manuscript. The author only summarized the main conclusions and did not give impressive implication.

**Reply:** Thanks for your comments. We added a section to emphase the implication:

[revised manuscript text omitted]

**Comments:** The author gave the results for F<=0.3. However, the aged BC particles have F ranges in 0-1 in the atmosphere. It seems the present results in this study are not complete. I think it is impossible to construct a shape model with large F and small coating thickness due to the limitation of the MSTM method (The coating must be sphere). If this is the situation, then why the author did not use DDA to calculate the results? If the data for F > 0.3 can not be supplemented, please clarify this in the main context.

**Reply:** Thanks for your comments. In this work, we just considered a spherical coating. Therefore, for fluffy BC, larger F may be not found. However, we expect that larger F may not modify the main conclusion of this work. In addition, for compact BC ($D_f =$

2.6), we have considered an F of 1 for comparison (please see Figure S1 – S2). In the future, more complex coating structures should be considered. We have clarified that in the revised manuscript:

"Since we consider only spherical coating structures in this work, a large F for completely BC may not be found for a BC volume fraction. Therefore, we only consider an F range from 0 to 0.3 for fluffy BC. However, we assume that BC with large F would not change the main results of this work."

**Specific**:

**Comments:** Line 6: The term $ABS_{BrC}$ was not well explained. It is not "the estimated BrC absorption". The $ABS_{BrC}$ seems to be a critical parameter to understand the whole manuscript. In the Abstract, the meaning of $ABS_{BrC}$ should be clearly explained to help readers to understand the results mentioned in the Abstract.

**Reply:** Thanks for pointing it out. We corrected it in the revised manuscript.

**Comments:** Line 9: The full name of "WDA" was not mentioned before using this abbreviation.

**Reply:** Thanks for your comments. We have given the definition in the revised manuscript.

**Comments:** Lines 181-182: "the corresponding $r_{max}$ and $r_{min}$ are 0.0342 μm and 0.2 μm" seems a wrong sequence for $r_{max}$ and $r_{min}$.

**Reply:** Thanks for pointing it out. We have corrected it in the revised manuscript.

**Comments:** Lines 235-237 and 238-239: It seems that the $AAE_{440\_870} = 1$ method often has larger $ABS_{BrC}$ than $AAE_{440\_675} = 1$ method. Why this happens? The results should be tried to explain here.

**Reply:** Thanks for your comments. The reason may the that the gap between 440 and 870nm is further than the gap between 440 and 675nm, which leads to larger $ABS_{BrC}$. We have clarified it in the revised manuscript:

"The $AAE_{40\_870} = 1$ method generally shows a larger range of $ABS_{BrC}$ than the $AAE_{440\_675} = 1$. This could be due to the larger wavelength gap between 440 and 870 nm. "

**Comments:** Lines 243-244: The sentence is too complicated. Maybe the author miss some punctuation marks.

**Reply:** Thanks for your comments. We have modified the sentence as "With $D_f$ of 1.8, F of 0.1, and $D_p/D_c$ of less than 2.71, $ABS_{BrC}$ varies in the range of about -6% - 18% and -12% - 9% when $AAE_{440\_675} = 1$ and $AAE_{440\_870} = 1$. The ranges become -18% - 3% and -21% - 4.\% when $D_p/D_c$ is 4.64.".

**Comments:** Line 246: "can be observed" I don't think "observed" is a suitable word here.

**Reply:** Thanks for your comments. We modified "observed" as "found" in the revised manuscript.

**Comments:** "On the other hand, the AAE increases with Dp/Dc when BC has a fluffy structure. Thus, the AAE can be greater than 1 when the fluffy BC is partially coated with a thick coating (Zhang et al., 2020b; Luo et al., 2023), resulting in $ABS_{BrC}$ of less than 0." The author tried to explain why the $ABS_{BrC} < 0$, but there is no direct causal relationship between the former and the latter sentence. The author needs to give more reasonable explanation.

**Reply:** Thanks for your comments. We are very sorry for without clarifying clearly the reason. As shown in the manuscript, the absorption coefficient of BC, which is incorrectly attributed to BrC, can be calculated as follows:

$$\Delta_{BrC} = b_{abs\_BC\_440\_Estmiated} - b_{abs\_BC\_440}$$

$$b_{abs\_BC\_440\_Estmiated} = b_{abs_{BC_\lambda}} \left(\frac{440}{\lambda}\right)^{-1}$$

$$b_{abs\_BC\_440} = b_{abs_{BC_\lambda}} \left(\frac{440}{\lambda}\right)^{-AAE\_true}$$

Thus, if the AAE of partially coated BC is greater than 1, as $\lambda > 440$, so $b_{abs\_BC\_440} > b_{abs\_BC\_440\_Estmiated}$. Therefore, $ABS_{BrC} < 0$. We have added clarifications in the revised manuscript:

"On the other hand, the AAE increases with Dp/Dc when BC has a fluffy structure. Thus, the AAE can be greater than 1 when the fluffy BC is partially coated with a thick coating (Zhang et al., 2020b; Luo et al., 2023). This would result in the predicted black carbon absorption coefficient being larger than the true black carbon absorption coefficient, so resulting in $ABS_{BrC}$ of less than 0."

**Comments:** Line 256: "The more compact structure can also represent another process of atmospheric aging". The compact structure was caused by many aging processes. I wonder the author means which process?

**Reply:** Thanks for your comments. We have re-written the sentences:

"With atmospheric aging, the BC cores are reconstructed to be a more compact structure. We used a larger Df ($D_f = 2.6$) to represent the compact BC. Even with $F = 0$, a Df of 2.6 represents the highly aged BC. By comparing BC with fluffy and compact structrues, we can see more deeply from the effects of atmospheric aging on the estimations of BrC absorption."

**Response to the comments of Reviewer #2**

**Comments:** The study aims to quantify how the microphysical properties of BC affect BrC absorption estimates, providing insight into the uncertainties surrounding the BrC assessment through absorption measurements at various wavelengths. This study is very useful and significant which may bring an important advance in the estimates of BrC absorption.

**Reply:** Thanks very much for your positive comments for our work. The response for the comments are show in the following.

**Comments:** Main: The results obtained in this work are impressive and significant, however, the presentation of these results should be better. Be more succinct and direct, highlight the main results, improve the figures (tables?), and worry about making better captions in the figures and not just explaining them in the middle of the text, this will make reading pleasure.

**Reply:** Thanks very much for your valuable suggestions. We have improved the presentations of the results in the revised manuscript. We have replotted Figure 3 – 4, and define the data in Tables 1 -5. Please see the revisions that marked in blue in the revised manuscript.

**Comments:** Below are some comments and specific suggested adjustments to the text.

**Reply:** Thanks very much for your comment. The response for the comments are show in the following.

**1 Introduction**

**Comments: line 22**- Carbonaceous aerosols are a major contributor to climate "change".

**Reply:** Thanks for pointing it out. We have corrected it in the revised manuscript.

**Comments: line 55** - A quick and effective way to differentiate BC, BrC, and dust using Scattering Angstrom exponent x Absorption Angstrom Exponent ratios is

described in studies such as (CAPPA et al., 2016; CAZORLA et al., 2013; RUSSELL et al., 2010).

**Reply:** Thanks for your suggestion for the references, and we have added them in the revised manuscript.

**Comments: line 70** – "while more recent studies based on measurements and simulations have shown a wide range of AAE values". Which studies? Please mention it here.

**Reply:** Thanks for your comments. We have added some references here.

**2 Estimating the BrC absorption**

**Comments: Lines 118-120** – "we assumed the following cases for aged BC aerosols: (1) fluffy BC cores partially coated with other materials; (2) compact BC without coating materials; (3) compact BC partially coated with other materials; (4) compact BC fully coated with other materials". In my opinion, they are very promising choices.

**Reply:** Thanks for your comments.

**Comments: Figure 1** – Please explain in more detail each of the 4 cases in the figure legend.

**Reply:** Thanks for your comments. We have added the explaination in the revised manuscript:

"We used a $D_f$ of 1.8 and 2.6 to represent fluffy BC and compact BC, respectively. Moreover, $F = 0$ means that BC is not internally mixed with coating materials, while BC is gradually partially coated with other materials as F increases. $F = 1$ means that BC is completely coated."

**Comments: Line 157** - of nonabsorbing

**Reply:** Thanks for your comments. We have corrected it in the revised manuscript.

**3 Results**

**Comments:** The results text is quite challenging to understand, with a lot of technical data and few graphics, and tables. It would be great to explain the results more pleasantly. Maybe doing better figures.

**Reply:** Thanks very much for your comments. We have added Tables 1- 5 to show the technical data, and we have also added contour lines to show the data in Figures 3 – 4.

**Comments:** Did you use AERONET data? which sites? make it clear and explicit in the text.

**Reply:** Thanks for your comments. We did not use AERONET data. The $ABS_{BrC}$ in this work is retrieved from the absorption of black carbon. Consequently, any deviations from brown carbon of $ABS_{BrC} = 0$ indicate uncertainty in the BrC estimation. This work is based on modeling only; no measurement data are used. We have clarified it in the revised manuscript:

"The $ABS_{BrC}$ in this work is retrieved from the absorption of black carbon. Consequently, any deviations from $ABS_{BrC} = 0$ suggest the uncertainty in BrC estimation. This work is based solely on modeling, and no measurements are used."

**Comments: Figure 2**. ABSBrC at different Dp/Dc estimated using the AAE = 1 methods. What different Dp, Dc and F indicate? Please clarify at legend too. Example: The increases in F may represent a process of atmospheric aging.

**Reply:** Thanks very much for your comments. We added the explaination in the caption:

"When freshly emitted, BC generally exhibits a flocculent structure and is not internally mixed with coating materials, as reflected by an F of 0 and a $D_f$ of 1.8. However, with increasing atmospheric aging, BC gradually becomes internally mixed with coating materials and thicker coating materials overlay on the BC, which may be reflected in a larger F and Dp/Dc. Moreover, the BC structure becomes more compact as the particles age, and we used a large Df value to represent the compact BC."

**Comments: Figure 3 – 4**, Add the explanation on the legend: "we see that $ABS_{BrC}$ increases with rg when AAE is fixed. This is caused by a decrease in AAE with increasing rg for fluffy BC". The article will be easier to read

**Reply:** Thanks very much for your suggestion. We have added the clarification in the revised manuscript.

**Comments: Figure 5**. "Similar to Figure 3, but using the WDA method". Please it

**Reply:** Thanks very much for pointing it out. The caption was modified as "Similar to Figure 2, but using the WDA method.We see that the WDA method does not always provide better estimates than the fixed AAE methods, and its applicability is also significantly affected by morphology and mixing states."

**Comments: Lines 303 – 304** "The estimated $ABS_{BrC}$ based on the WDA using the bare sphere model and the core-shell model has comparable values and can vary in a range from about -40% to 36%". What does it mean? Explain.

**Reply:** Thanks very much for your comments. We have modified the sentence as:" The estimated $ABS_{BrC}$ based on the Mie theory-based WDA can vary in a range from about -40.8% to 35.7%"

**Comments: Figure 6** - Please explain the figure better, what does this purple bar mean for example? What significant result did you get? highlight this in the caption.

**Reply:** Thanks very much for your valuable comments. The purple shading represents the WDA calculated for the bare spherical BC model. We have added the finding in the caption:

"The results show that the WDA of BC is significantly affected by morphology and mixture state. When BC is freshly emitted (represented by a $D_f$ of 1.8, an F of 0, and a smaller $D_p/D_c$), the WDA is close to 0. With atmospheric aging, the WDA range becomes broader and often deviates significantly from 0. In addition, the WDA

calculated using the Mie assumption can differ greatly from the WDA of partially coated BC, which is why the $ABS_{BrC}$ estimated using the WDA method differs from 0."

**Comments: Lines 388-389** "The estimated DRF in this work may be about +0.216 – +0.612 W/m2 , which is generally in the range of values reported by previous studies". How did you estimate it?

**Reply:** The DRF is estimated based on the suggested values of Bond et al. (2013). From the Table 15 of Bond et al. (2013), Kelesidis et al. (2022) estimated an average absorption forcing efficiency of $170\pm43$ $Wm^{-2}$ /AAOD for BC used in global climate models, please see Table 15 in Bond et al. (2013) and Kelesidis et al. (2022). In this work, we estimated the DRF by multiplying the estimated AAOD with $170\pm43$ $Wm^{-2}$ /AAOD. Similarly, the misassigned DRF is also estimated by multiplying the misassigned AAOD by $170\pm43Wm^{-2}$ /AAOD. We have clarified it in the revised manuscript:

"In addition to AAOD, the Direct Radiative Forcing (DRF) is also commonly used to assess climate effects. In this work, DRF is also estimated using a simple method. Based on the values in Bond et al. (2013), Kelesidis et al. (2022) proposed to use an average absorption forcing efficiency of $170 \pm 43$ $Wm^{-2}$ /AAOD. Similar to Kelesidis et al. (2022), the DRF is estimated by multiplying the estimated AAOD by $170 \pm 43$ $Wm^{-2}$ /AAOD. Similarly, the misassigned DRF is also estimated by multiplying the mis-assigned AAOD by $170\pm43Wm^{-2}$ /AAOD."

**Comments: Figure 9.** "The global mean BC AAOD calculated using different models". What is the most important result of this figure? explain. Perhaps this is one of the most important figures in his work.

**Reply:** Thanks very much for your comments. We have modified the caption as "The global mean BC AAOD was calculated using different models. Our case studies show a global mean AAOD value of about 0.0016 to 0.0026, which is significantly affected by mixing states and morphology. The AAOD of BC with a fluffy morphology is

generally larger than that of BC with a compact morphology. Moreover, the predicted AAOD does not necessarily increase with increasing coating ratios (decreasing $f_{BC}$), which is due to the shielding effects of the coating materials. Moreover, the AAOD of BC with thicker coating materials (smaller fBC) is more sensitive to the size distribution."

**4 Conclusions and Summary**

**Comments:** In conclusion, try to find explanations for the results obtained and not just repeat succinctly what was said in the results.

**Reply:** Thanks for your comments. We have added a section to give the implication:

"AAE-based methods have been widely used to estimate the absorption of BrC, while they are subject to large uncertainties due to the properties of BC. We quantify the effects of the microphysical properties of BC based on numerical simulations and investigate how the applicability of AAE-based methods varies at different aging conditions. From the above, it is clear that using a BC AAE of 1 can provide reasonable estimates for BrC absorption, while the deviation from the "true" BrC absorption becomes significant as the particles age. This means that the AAE = 1 method can provide inaccurate estimates when aged BC is present. In general, regions near emission sources, such as urban traffic areas, contain mainly freshly emitted BC. In this case, it is reasonable to use the AAE = 1 method. With atmospheric aging, we should adjust the AAE values because both the AAE = 1 and WDA methods can sometimes result in misallocations of tens of percent of BrC absorption. However, the adjustments should differ depending on the aging condition. As shown in Figure 3-4, for fluffy BC partially mixed with coating materials ($D_f = 1.8$ and $0 < F < 1$ in this work), $ABS_{BrC} = 0$ occurs in most cases when AAE > 1. Therefore, we generally propose a relatively larger AAE, while a smaller AAE is recommended for compact BCs, including coated and uncoated compact BCs. Recent observations have shown that the average $D_f$ is often small even for coated BCs in regions far from emission sources. Therefore, we

prefer larger AAEs. However, there are also more compact BCs in the atmosphere, and we should also consider the uncertainties when the real BCs have a compact structure."

Besides, we have also rewritten the "Conclusion and summary":

"The AAE-based method is commonly used to estimate the absorption of BrC, but may provide inaccurate estimates due to the effects of the microphysical properties of BC. The goal of this work is not to discuss the use of the AAE-based method, but to assess the uncertainties of the AAE-based method. We find that an AAE of 1 can provide a reasonable estimate when BC is freshly emitted. Therefore, an AAE of 1 is suggested for regions close to the emission source, such as vehicle emission region. However, we should also note the uncertainties associated with using an AAE of 1. We estimate an $ABS_{BrC}$ range of about -4.8% to 2.7% when using an AAE of 1 for freshly emitted BC. However, the $ABS_{BrC}$ range becomes broader when BC is aged, and sometimes $ABS_{BrC}$ can be varied in the range of about -34.5% – 38.7%, depending on the aging status and morphologies. Therefore, we need to adjust the AAE value when the fixed AAE method is applicable to the region consisting of aged BC, such as regions far from the emission source. However, even for aged BC, different AAE values should be used for different aging conditions, since we show that no fixed AAE is applicable for all cases.

This work represents the aging condition by assuming a more compact structure, more coating materials and a larger F. For different aging processes, the adjustment of AAE values should be different. For fluffy BC partially mixed with coating materials (Df = 1.8 and 0<F<1 in this work), we generally propose a larger AAE, while a smaller AAE is recommended for the compact BC. Our results also show that the Mie theory-based WDA method does not necessarily improve the estimate, with a corresponding ABSBrC range of about -40.8% – 35.7% in our simulation cases, due to the substantial WDA deviation between the morphologically realistic BC and the spherical BC.

At the global level, the use of BC AAE of 1 can lead to a global mean misassigned AAOD of about -0.43 – $0.46 \times 10^{-3}$ resulting in a corresponding global mean misassigned DRF of -0.073 ± 0.0185 to +0.078 ± 0.0198 W/m2. However, for the freshly emitted BC, an AAE of 1 does not lead to a significant misestimation of the AAOD. At the regional level, for an AAE of 1, the mean mis-assigned AAOD can vary in the range of -7.3 to $5.7 \times 10^{-3}$ in some regions, leading to a mis-assigned DRF of about -1.24 ± 0.314 W/m$^2$ to +0.97 ± 0.245 W/m$^2$. The WDA method can provide a less accurate estimate for BrC absorption, and sometimes in some regions we can see a mean mis-assigned AAOD of about $-22 \times 10^{-3}$, leading toa mis-assigned DRF of about -3.74 ± 0.946 W/m$^2$. Therefore, the effects of the microscopic properties of BC should be carefully considered when estimating BrC absorption and its direct radiative forcing based on the measurements at multiple wavelengths."

**Comments:** Will you prepare any material supplementary?

**Reply:** Thanks very much for your question. We have provided supplementary material in attached.

**Comments:** In summary, the work is very good and could mean a great scientific advance in BrC estimates from BC measurements. In addition, it can bring significant advances in the estimation of the radiative forcing of absorbing aerosols. Congratulations on a great job.

**Reply:** Thanks very much for your positive evaluation for our work!